# Long-term ex situ normothermic perfusion of human split livers for more than 1 week

Ngee-Soon Lau [1,2,3], Mark Ly[1,2,3], Claude Dennis[4], Andrew Jacques[1,2,3], Marti Cabanes-Creus[5], Shamus Toomath[1,2], Joanna Huang [1,2,3], Nicole Mestrovic[1,2,3], Paul Yousif[1,2], Sumon Chanda[1,2], Chuanmin Wang[1,2,3], Leszek Lisowski [5,6,7], Ken Liu[2,3], James G. Kench[3,4], Geoffrey McCaughan[1,2,3,8], Michael Crawford[1,2,3] & Carlo Pulitano [1,2,3] ✉

Current machine perfusion technology permits livers to be preserved ex situ for short periods to assess viability prior to transplant. Long-term normothermic perfusion of livers is an emerging field with tremendous potential for the assessment, recovery, and modification of organs. In this study, we aimed to develop a long-term model of ex situ perfusion including a surgical split and simultaneous perfusion of both partial organs. Human livers declined for transplantation were perfused using a red blood cell-based perfusate under normothermic conditions (36 °C) and then split and simultaneously perfused on separate machines. Ten human livers were split, resulting in 20 partial livers. The median ex situ viability was 125 h, and the median ex situ survival was 165 h. Long-term survival was demonstrated by lactate clearance, bile production, Factor-V production, and storage of adenosine triphosphate. Here, we report the long-term ex situ perfusion of human livers and demonstrate the ability to split and perfuse these organs using a standardised protocol.

Normothermic machine perfusion technology offers a number of advantages over traditional techniques for the preservation of organs prior to transplantation[1]. Perfusing a donated human liver prior to transplant can extend the ex situ preservation time in the short term and simultaneously allow some assessment of organ viability as a predictor of post-transplant graft function[2,3]. The primary focus of this technology to date has been to increase the utility of marginal organs using short-term perfusion in the range of hours. However, perfusion in the range of days to weeks could facilitate a more sophisticated assessment of these organs with the potential for recovery or mod-

ification prior to transplant[4,5]. Not only could this increase the number of available organs for transplantation, but it could also improve the quality of the grafts currently used.

To this end, perfusion of livers for up to 7 days has been reported using a custom-built, integrated system under subnormothermic conditions (34 °C)[4]. Perfusion at this temperature has protective metabolic effects but does not simulate actual physiological conditions[6,7]. The same group also reported the successful transplant and 1-year follow-up of a liver that was perfused using ex situ normothermic preservation for 3 days[8]. Long-term perfusion of human

[1]Centre for Organ Assessment Repair and Optimisation, Royal Prince Alfred Hospital, Sydney, New South Wales 2050, Australia. [2]Australian National Liver Transplantation Unit, Royal Prince Alfred Hospital, Sydney, New South Wales 2050, Australia. [3]Faculty of Medicine and Health, The University of Sydney, Sydney, New South Wales 2006, Australia. [4]Department of Tissue Pathology and Diagnostic Oncology, NSW Health Pathology, Royal Prince Alfred Hospital, Sydney, New South Wales 2006, Australia. [5]Translational Vectorology Research Unit, Children's Medical Research Institute, The University of Sydney, Westmead, Sydney, New South Wales 2145, Australia. [6]Military Institute of Medicine, Laboratory of Molecular Oncology and Innovative Therapies, 04-141 Warsaw, Poland. [7]Australian Genome Therapeutics Centre, Children's Medical Research Institute and Sydney Children's Hospitals Network, Westmead, NSW 2145, Australia. [8]Centenary Institute, Sydney, New South Wales, Australia. ✉e-mail: carlo.pulitano@sydney.edu.au

livers beyond 7 days using normothermic conditions (36 °C) has never been reported and could unlock the potential for regeneration and modification of organs before transplant.

Long-term ex situ perfusion of human livers using normothermic conditions also represents a unique opportunity for studying living human tissue ex situ. By splitting whole human livers during normothermic machine perfusion as we have previously described[9–11], this technology can be applied to two partial livers. This could provide a simulated environment for the testing of therapeutics with a matched control and the study of liver injury and regeneration.

In this study, we aimed to develop a proof-of-concept model of long-term normothermic ex situ perfusion of human split livers to push the boundaries of ex situ perfusion by extending survival beyond 7 days and simultaneously perfusing two partial organs. In this way, we sought to develop a model to investigate long-term liver perfusion with potential applications in translational research and beyond.

## Results

### Donor demographics

All donor livers in New South Wales consented for research and declined for clinical transplantation between February and December 2021 were considered for inclusion. One liver was declined due to a known history of portal hypertension, and a second due to cirrhosis. To develop the protocol, three whole livers were perfused without splitting. Using our protocol, 10 donated human livers were split, resulting in 10 LLSG and 10 ERGs which were perfused on separate perfusion machines.

For the livers that were split, four of the ten livers were procured through the donation after brain death (DBD) pathway, and six were procured through the donation after circulatory determination of death (DCD) pathway. The reasons for discard of donated livers are summarised in Table 1. Some of these livers may have been "transplantable" in other centres, but the decision to use them was not influenced by our study protocol. The reasons for not using the DBD livers for transplantation were: steatosis, severely deranged liver function tests, known biliary sepsis, and a liver with a poor back-table flush. The DCD livers were not used clinically due to age beyond DCD acceptance criteria at our centre (>50 years) in 3/6, and the remaining due to a prolonged time to the cessation of circulation (>30 min), morbid obesity, and acuity of transplant activity. The median cold ischaemic time (CIT, defined as the time from cold perfusion to reperfusion using the ex situ machine) was 295 min (interquartile range [IQR] 273–430 min) (Supplementary Table 1). For DCD livers, the median time to death (withdrawal of cardiorespiratory support to cessation of circulation) was 20 min (IQR 19–29 min) (Supplementary Table 1).

### Whole liver perfusion

All whole livers (3/3) demonstrated rapid clearance of lactate following the commencement of perfusion (Supplementary Fig. 1A). Using our protocol for long-term perfusion, these whole livers all survived for >7 days with evidence of lactate clearance and bile production (Supplementary Fig. 1B). Once lactate started to rise beyond 2.5 mmol/L, we observed an irreversible deterioration in organ function which ultimately ended in organ failure in all cases.

### Organ viability and survival

All livers were continuously assessed for hepatocellular viability according to the criteria proposed by the VITTAL clinical trial[2]. Apart from two livers that failed early due to a technical error in organ transfer, all livers remained viable to 48 h of perfusion. At 5 days of perfusion, 11/20 livers continued to meet viability criteria, and the median time of viability overall was 125 h (IQR 88–176 h) (Fig. 1A, Supplementary Table 2). After lactate increased to >2.5 mmol/L and viability criteria were no longer fulfilled, perfusion was continued for all partial livers in an exploratory fashion to characterise changes relating to organ failure. The time from being non-viable to complete organ failure (lactate >10 mmol/L and exponentially rising with a lack of bile production or unresponsive hypoglycaemia) was typically <48 h (16/20 grafts). The overall median survival was 165 h (IQR 113–224 h), with 9/20 livers surviving for >7 days and 4/20 livers surviving >10 days (Fig. 1B, Supplementary Table 2). The maximum overall survival was 327.5 h. Hepatobiliary viability was assessed using criteria from the DHOPE-COR-NMP trial[12]. The same two livers that failed due to a technical error were also not viable by these criteria, but all other partial livers met these hepatobiliary viability criteria for up to 48 h of perfusion (Supplementary Table 3). Notably, these livers also all produced bile with a pH >7.40, indicating preserved cholangiocyte function (Supplementary Table 3).

### Liver biochemistry

All partial livers demonstrated clearance of lactate. Residual lactate from packed red blood cells used to synthesise the perfusate was rapidly cleared by the whole liver, and then the ERG after splitting (the ERG was transferred to the second perfusion machine with "new" blood, while the LLSG remained connected to the first perfusion machine). Lactate clearance continued during perfusion with the maintenance of a perfusate lactate level typically <1.5 mmol/L until the

## Table 1 | Reasons for discard of donated livers

| Liver number | Blood group | Donor type | Cause of death | Reason for discard |
|---|---|---|---|---|
| 1 | A | DBD | Cardiac arrest, asthma | 30–60% steatosis on biopsy |
| 2 | B | DCD | Traumatic brain injury | DCD*, age |
| 3 | O | DCD | Traumatic brain injury | DCD, time to cessation of circulation >30 min |
| 4 | B | DCD | Intracranial haemorrhage | DCD, age |
| 5 | O | DBD | Hypoxia, cardiac arrest | Medically unsuitable, biliary sepsis, cholecystostomy |
| 6 | A | DBD | Cerebral oedema, sepsis, multiorgan failure | Deranged LFTs |
| 7 | O | DCD | Hypoxia, cardiac arrest, aspiration | DCD, age, time to cessation of circulation >30 min |
| 8 | O | DCD | Hypoxia, respiratory failure requiring veno-venous extra-corporeal membrane oxygenation | DCD, weight, comorbidities |
| 9 | A1 | DCD | Hypoxia, cardiac arrest. | DCD, acuity of transplant activity, concurrent DBD donor |
| 10 | O | DBD | Intracranial haemorrhage | Poor flush, donor surgeon assessed as not suitable for transplantation |

*Criteria for acceptance of DCD organs: Age <50, time to cessation of circulation <30 min.
*DBD* donation after brain death, *DCD* donation after circulatory determination of death.

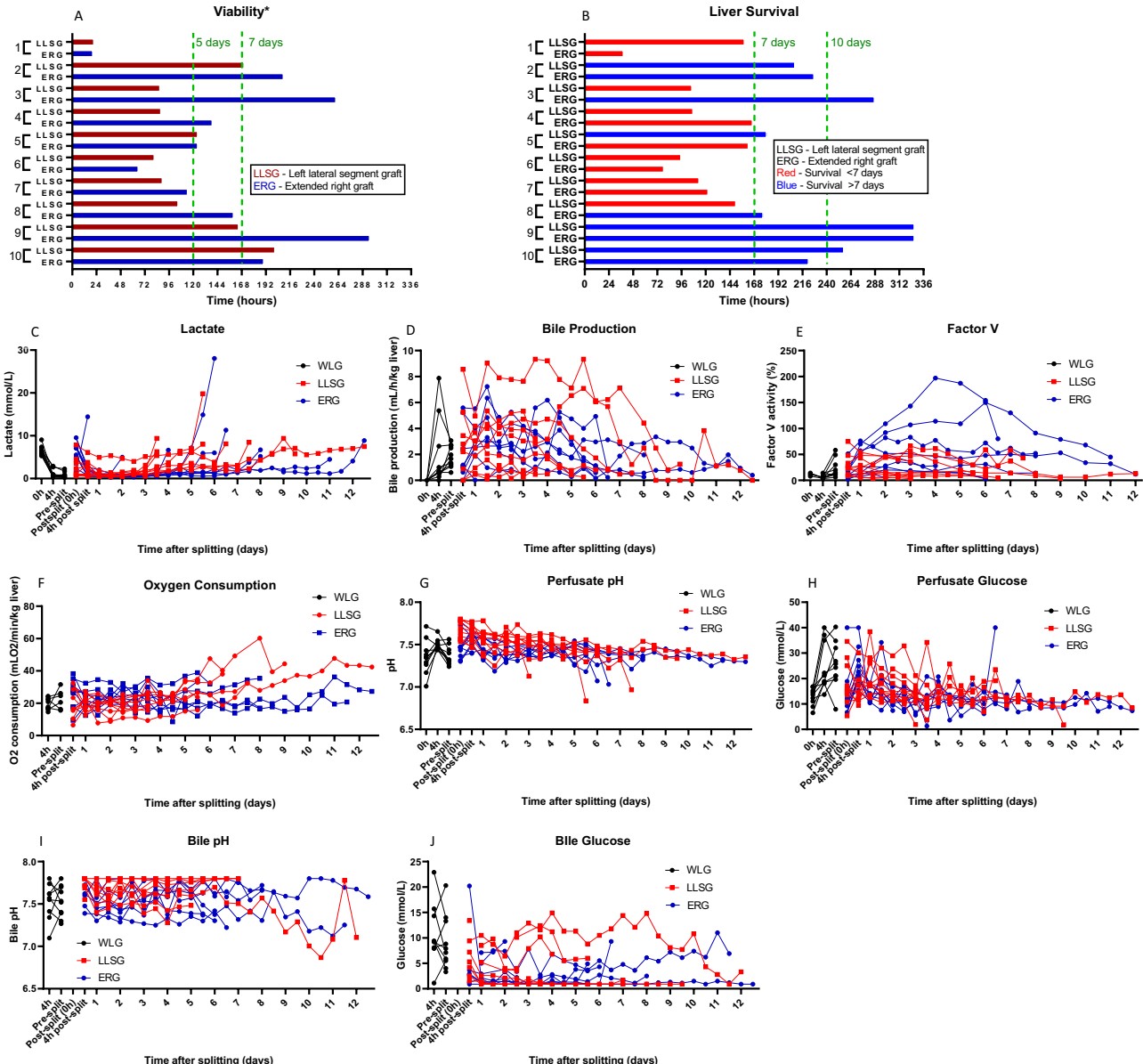

**Fig. 1 | Biochemical and functional evidence of the long-term function of human split livers.** Organ viability was continuously assessed until partial livers no longer fulfilled viability criteria* (**A**). Perfusion continued until organs failed (**B**), characterised by a lactate >10 mmol/L with a lack of bile production or unresponsive hypoglycaemia. All livers demonstrated lactate clearance (**C**), bile production (**D**), production of Factor-V (**E**), and evidence of oxygen consumption (**F**) until the point of organ failure. Perfusate pH and glucose were typically stable during perfusion until organ failure, which resulted in refractory acidosis and unresponsive hypoglycaemia (**G**, **H**). Bile pH was typically alkalotic and bile glucose was typically in the hypoglycaemic range during perfusion (**I**, **J**). *Viability according to the criteria proposed by the VITTAL clinical trial (≤2.5 mmol/L, and two or more of: bile production, pH ≥ 7.30, glucose metabolism, hepatic arterial flow ≥150 ml/min and portal vein flow ≥500 ml/min, or homogeneous perfusion)[2].

point of organ failure, where the lactate increased exponentially (Fig. 1C). Bilirubin, GGT, and ALP levels in the perfusate remained low but gradually increased during perfusion towards the point of organ failure (Supplementary Fig. 2). Perfusate levels of ALT peaked in the first 48–72 h after whole liver reperfusion before plateauing then decreasing during long-term perfusion (Supplementary Fig. 2).

## Liver function
Bile production was constant throughout perfusion in both left and right livers, with comparable rates of production after adjustment to the weight of each partial liver. Production typically slowed and ceased corresponding with organ failure (Fig. 1D). Bile pH was typically maintained in an alkalotic range during perfusion (Fig. 1I). Similarly,

perfusate prothrombin time (PT) and levels of Factor-V were maintained during perfusion in all partial livers until organ failure, and there was evidence of oxygen consumption in all livers in the long-term (Fig. 1E, F, Supplementary Fig. 2D). The perfusate pH was typically stable between 7.2 and 7.5 during perfusion, with a characteristic refractory acidosis that corresponded with the final stages of organ failure (Fig. 1G). The perfusate levels of albumin, urea and total protein were maintained throughout perfusion without noticeable change at organ failure (Supplementary Fig. 2).

## Glucose
The perfusate glucose level was typically high in the first 48–72 h of perfusion and subsequently maintained in the range of 10–20 mmol/L

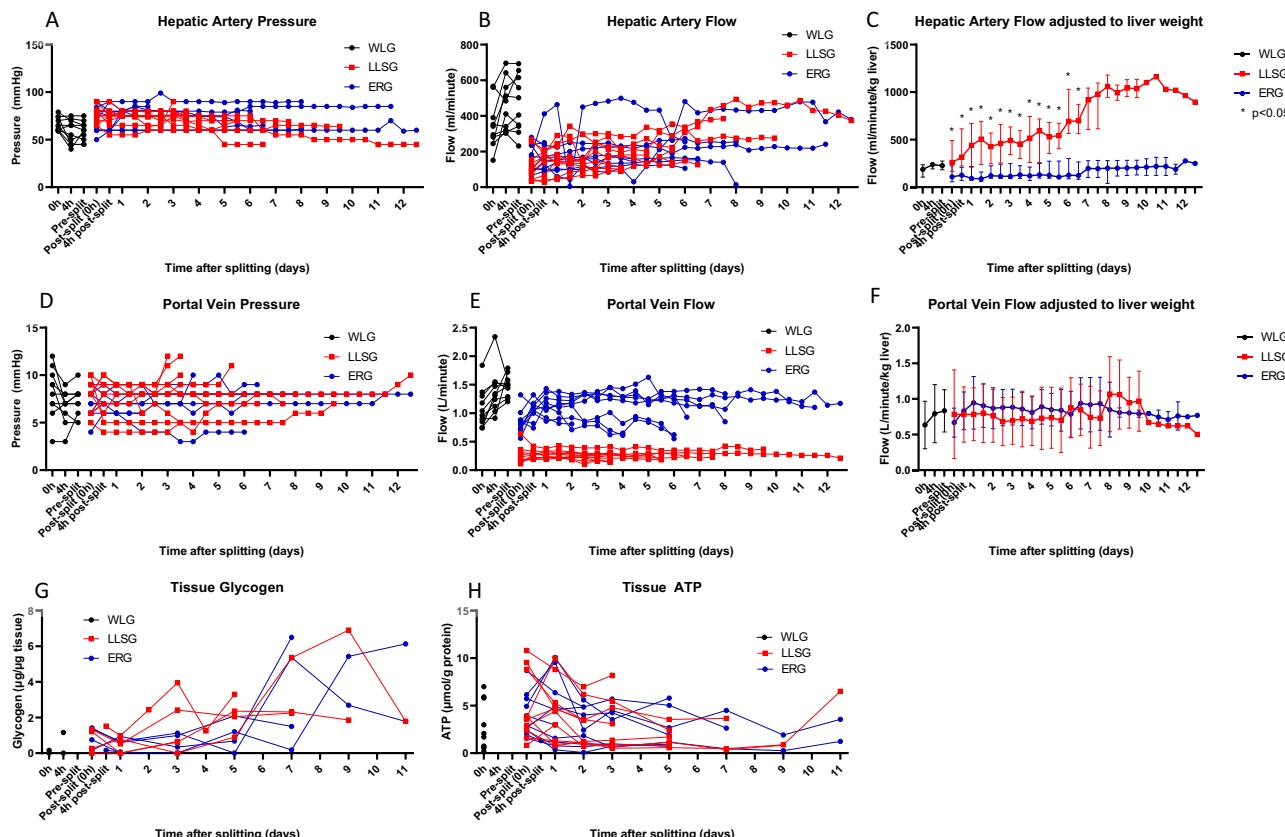

**Fig. 2 | Monitored haemodynamic indices and tissue ATP and glycogen during long-term perfusion of human split livers.** Using a pressure-controlled system with a goal of 60 mmHg for the hepatic artery, the ERG and LLSG typically achieved 200 ml/min of blood flow (**A**, **B**). After adjustment for liver weight, the LLSG achieved significantly higher flows/min/kg of liver than the ERG (median 316 ml/min [IQR 224–613 ml/min] vs 126 ml/min [IQR 72–209 ml/min], $p = 0.003$, at 4 h after splitting, Mann–Whitney U Test) (**C**) ($n = 20$ partial livers; 10 ERGs, 10 LLSGs, expressed as median (IQR)). With a goal of 8 mmHg for the portal vein, the ERG and LLSG typically achieved 1.0 L/min and 300–400 ml/min, respectively (**D**, **E**). After adjustment for liver weight, the portal vein flow was similar between ERG and LLSGs (**F**) ($n = 20$ partial livers; 10 ERGs, 10 LLSGs, expressed as median (IQR)). Hepatic tissue ATP and glycogen levels remained stable or increased during perfusion in both ERGs and LLSGs compared to baseline (**G**, **H**). ATP adenosine triphosphate, ERG extended right graft, LLSG left lateral segment graft, *$p < 0.05$.

during long-term perfusion (Fig. 1H). The end of perfusion was characterised by refractory hypoglycaemia corresponding with organ failure.

## Vascular haemodynamics

We utilised a pressure-controlled system that permitted independent control of hepatic arterial and portal venous pressure with measurement of vascular flow. In the ERG, cannulation of the right hepatic artery typically achieved flows of 200 ml/min, and in the LLSG, cannulation of the coeliac trunk typically also achieved flows of 200 ml/min (Fig. 2A, B). After adjustment for liver weight, the LLSG achieved significantly higher flows/min/kg of liver than the ERG (median 316 ml/min [IQR 224–613 ml/min] vs 126 ml/min [IQR 72–209 ml/min], $p = 0.003$, at 4 h after splitting) (Fig. 2C). For the ERG, cannulation of the main portal vein typically achieved flows of 1.0 L/min, while in the LLSG, cannulation of the left portal vein achieved only 300–400 ml/min (Fig. 2D, E). Adjustment by liver weight demonstrated that this flow rate was similar between ERGs and LLSGs (Fig. 2F). Vascular pressures and flows typically remained stable throughout perfusion without vasodilator supplementation. Organ failure was characterised by a sharp increase in hepatic arterial resistance immediately before perfusion was terminated.

## Tissue adenosine triphosphate and glycogen

Liver core biopsies taken throughout perfusion showed evidence of energy storage during long-term perfusion. Hepatic tissue levels of adenosine triphosphate (ATP) and glycogen remained stable or increased during perfusion in both LLSGs and ERGs compared to baseline (Fig. 2G, H).

## Histopathology

Liver architecture was preserved in the long term for both LLSGs and ERGs with low rates of coagulative necrosis or hepatocyte detachment during perfusion (Fig. 3). Increasing evidence of liver injury and architectural distortion was noted in the final stages of organ failure (Fig. 3). Glycogen depletion decreased during perfusion, demonstrating increased deposition of intracellular glycogen within hepatocytes in the long term (Fig. 3B). Evidence of cellular proliferation was noted, with hepatocytes staining positive for ki67 before and after splitting. However, the amount of cellular proliferation decreased towards the end of perfusion (Fig. 3C, D). Biliary injury was present in bile duct biopsies with evidence of mural stromal necrosis; however, 80% of livers demonstrated intact biliary epithelium suggesting preserved biliary tree integrity (Supplementary Fig. 3).

## Factors related to long-term survival

By grouping those partial livers that survived >7 days or ≤7 days, we examined the factors that predicted long-term survival. In total, 9/20 partial livers survived >7 days. This included 4 LLSGs and 5 ERGs, and these partial livers were derived from six different whole livers. Donor characteristics were not significantly different between the two groups. The mean donor age for livers that survived >7 days and

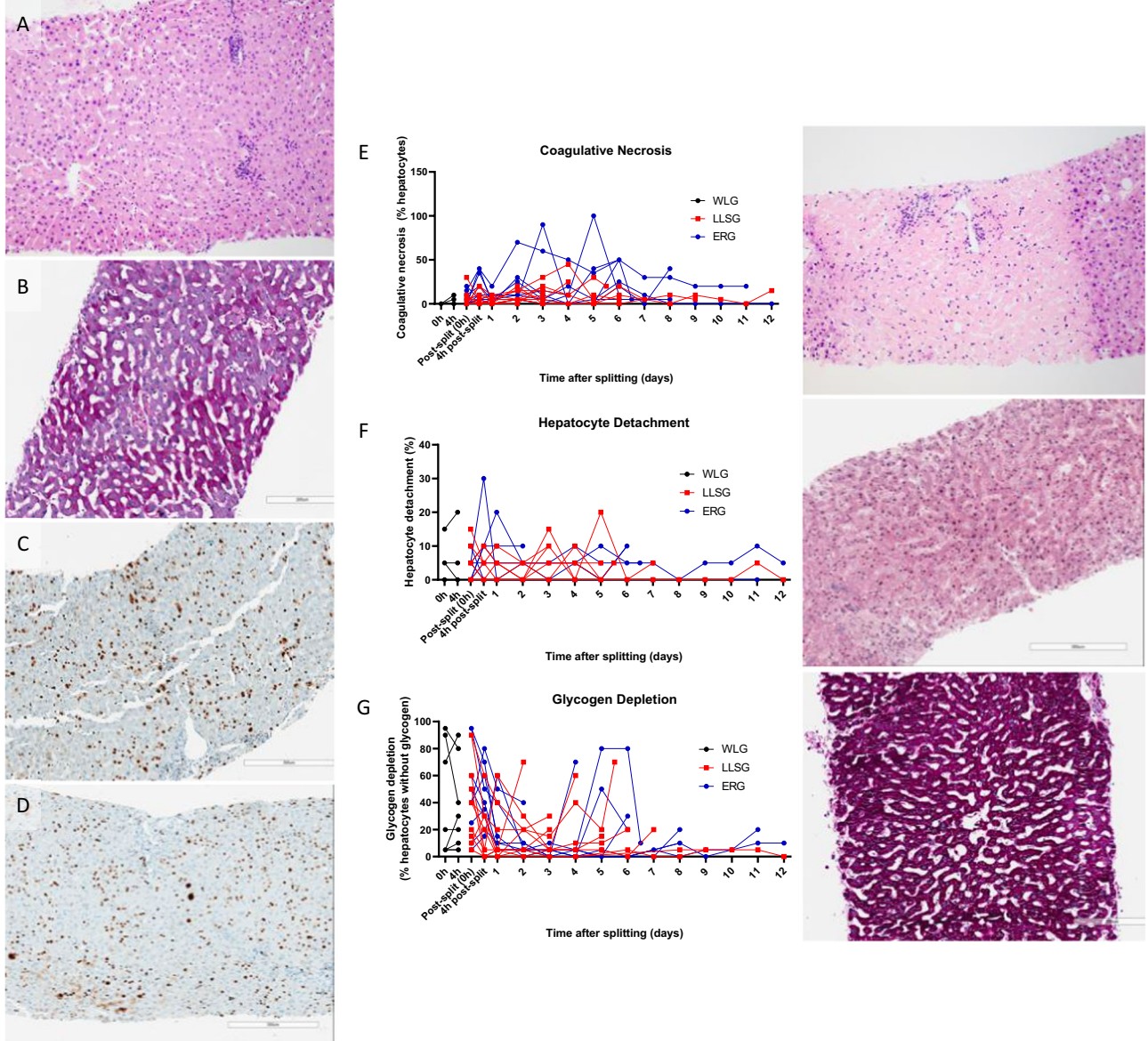

**Fig. 3 | Histopathology analysis of liver core biopsies taken throughout long-term perfusion of human split livers.** Slides were stained with haematoxylin and eosin to assess architectural integrity (**A**), Periodic acid-Schiff for glycogen depletion (**B**) and ki67 for cellular proliferation (**C**, **D**). The amount of cellular proliferation decreased from early in perfusion (**C**) to later in perfusion (**D**). Assessment of each slide was performed by a blinded specialist pathologist for coagulative necrosis (**E**), hepatocyte detachment (**F**) and glycogen depletion (**G**)[22]. Levels of coagulative necrosis and hepatocyte detachment remained low until the point of organ failure (**E**, **F**). Glycogen deposition within hepatocytes increased with long-term perfusion (**G**).

≤7 days was 52.8 ± 13.3 and 53.6 ± 15.4 ($p$ = 0.908), respectively. Donors for all organs were more commonly male (7/9 for livers surviving >7 days and 7/11 for livers surviving ≤7 days) and more commonly retrieved through the DCD pathway (6/9 vs 6/11 respectively) (Supplementary Table 4).

Perfusate lactate and perfusate levels of transaminases (bilirubin, alanine aminotransferase [ALT], alkaline phosphatase [ALP], gamma-glutamyl transferase [GGT]) were not significantly different between the two groups (Fig. 4A, Supplementary Fig. 4). The weight-adjusted rate of bile production was significantly higher in the livers that survived >7 days at 24 h, 60 h and 72 h after splitting (median 3.674 ml/h/kg liver [IQR 2.247–4.576 ml/h/kg liver] vs 1.714 ml/h/kg liver [IQR 0.478–2.516 ml/h/kg liver], $p$ = 0.008 at 24 h) (Fig. 4B). The perfusate level of Factor-V was significantly higher in the livers that survived >7 days immediately before splitting and at every time point up until 72 h after splitting (mean 47.3 ± 19.9% vs 15.4 ± 12.7%, $p$ < 0.001 at 24 h)

(Fig. 4C). Perfusate PT was significantly shorter in livers that survived >7 days immediately before splitting and 4 h after splitting (Fig. 4D). Perfusate urea, albumin, total protein, bile pH, and bile glucose did not demonstrate significant differences between the two groups (Fig. 4, Supplementary Fig. 4).

Hepatic artery flow was significantly higher for those livers that survived >7 days both before and after splitting (median 615 ml/min [IQR 530–674 ml/min] vs 342 ml/min [IQR 308–405 ml/min], $p$ = 0.002, just before splitting) (Fig. 4J). This difference was evident using pressure control targets that were only modified to meet minimum flow requirements. After adjusting for the weight of each liver, this difference was still present but less pronounced (Supplementary Fig. 4). The portal venous flows were significantly higher for livers that survived >7 days between days 1 and 3 after splitting (median 1.030 ml/min [IQR 0.320–1.310 ml/min] vs 0.280 ml/min [IQR 0.220–0.970 ml/min], $p$ = 0.049, 1 day after splitting) (Fig. 4L).

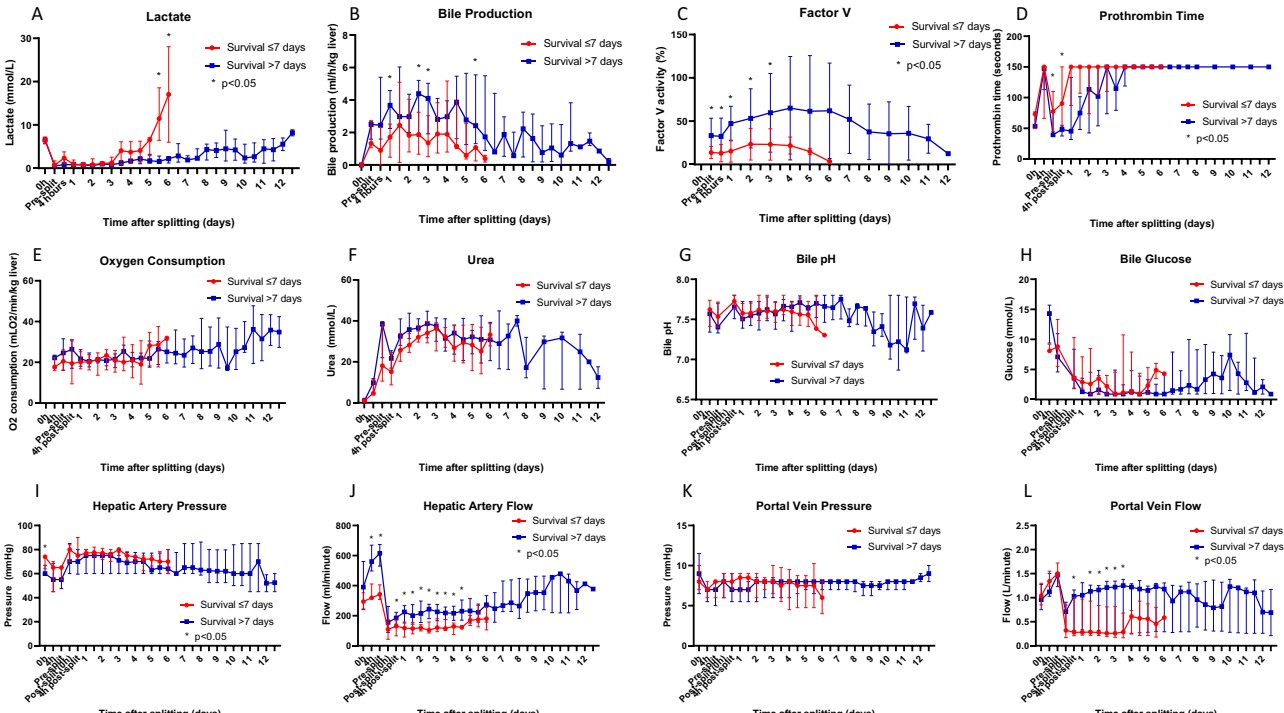

**Fig. 4 | Factors related to the long-term survival of human split livers.** Perfusate lactate levels were not significantly different between livers that survived >7 days or ≤7 days (**A**). Bile production and Factor-V levels were significantly higher in the livers that survived >7 days (bile: median 3.674 ml/h/kg liver [IQR 2.247–4.576 ml/ kg liver] vs 1.714 ml/h/kg liver [IQR 0.478–2.516 ml/h/kg liver], *p* = 0.008 at 24 h, Mann–Whitney U Test; Factor-V: mean 47.3 ± 19.9% vs 15.4 ± 12.7%, *p* < 0.001 at 24 h, unpaired two-sided *t*-test) (**B**, **C**). Prothrombin time was significantly shorter for livers that survived >7 days immediately before and 4 h after splitting (median 54 s [IQR 38–48 s] vs 150 s [IQR 55–91 s] at 4 h, *p* = 0.015, Mann–Whitney U Test) (**D**). Oxygen consumption, perfusate urea, bile pH and bile glucose did not demonstrate significant differences between the two groups (**E**–**H**). Hepatic artery flow was significantly higher in the livers that survived >7 days for the same hepatic artery

pressure (median 615 ml/min [IQR 530–674 ml/min] vs 342 ml/min [IQR 308–405 ml/min], *p* = 0.002, just before splitting, Mann–Whitney U Test) (**I**, **J**). Portal venous pressure was not significantly different between the two groups (**K**). Portal venous flow was significantly higher in the livers that survived >7 days between days 1–3 after splitting (median 1.030 ml/min [IQR 0.320–1.310 ml/min] vs 0.280 ml/min [IQR 0.220–0.970 ml/min], *p* = 0.049, 1 day after splitting, Mann–Whitney U Test) (**L**). All grouped data are presented as median (IQR) except for Factor-V, which was normally distributed and presented as mean (standard deviation), *n* = 20 partial livers, 9 survived >7 days, 11 survived ≤7 days. Normally distributed data and non-normally distributed data were compared at each grouped time point using an unpaired two-sided *t*-test and a Mann–Whitney U Test, respectively. *\*p* < 0.05.

The severity of microvesicular steatosis seen on core biopsies taken before splitting was significantly less in livers that survived >7 days (median 5% [IQR 0–7.5%] vs 20% [IQR 5–35%], *p* = 0.041 at 0 h) (Fig. 5A). However, the severity of macrovesicular steatosis, coagulative necrosis, and hepatocyte detachment was not significantly different between the two groups (Figs. 3E, F and 5B).

## Discussion

This is the first study to demonstrate a functional model of long-term normothermic machine perfusion of human split livers. Using livers that were not usable at our centre for transplantation, ex situ preservation was achieved for longer than 7 days and up to a maximum of 13 days. Not only is this the longest-ever reported ex situ preservation of human livers, but we also report the perfusion of two partial organs from the same whole liver. Perfusion in the range of days to weeks may facilitate more sophisticated viability assessment, repair of organs prior to transplantation, and even regeneration[1,4,5]. Therefore, this model has enormous potential for studying therapeutics with an ideally matched control and as a model for translational research. Although we chose to perform a traditional split to divide the liver into an LLSG and ERG for this study, our protocol could be easily adapted to full-left full-right splitting to provide two partial livers with even closer matching.

In addition, this has been achieved by modification of a commercially available liver perfusion system with readily available components (long-term oxygenators, gas blender, dialysis filter). This has

the advantage of being easily reproducible and accessible to other centres with an interest in long-term perfusion of whole or partial livers and opens the way for global collaboration and innumerable possibilities as a model for translational research.

This is also the first description of long-term perfusion of a paediatric-sized graft (LLSG). This has its own challenges relating to vessel kinking, maintenance of a physiological environment with an artificially high vascular flow and blood volume, and titration of nutritional requirements. Indeed, the high arterial and low portovenous flows relative to the ERG were most likely due to the use of the large coeliac trunk and small left portal vein for cannulation of the LLSG. However, 4/9 partial livers that survived >7 days were LLSGs. The machine perfusion revolution has yet to be realised in the field of paediatrics[13], perhaps due to technical challenges. Still, the adaptations and modifications achieved in this study pave the way for these advances.

Other challenges using this protocol included the intensity required for manual adjustment of infusions and dialysis settings. The modified commercial system utilised in this study always required close supervision and algorithmic adjustment of parameters (Supplementary Fig. 5). This could be improved in future by digital control and automation of the machine using a customised design and integration of multiple systems.

Long-term machine perfusion of the liver remains largely unstudied. A Swiss group successfully perfused whole livers and partial livers after hepatectomy using a custom-built integrated perfusion

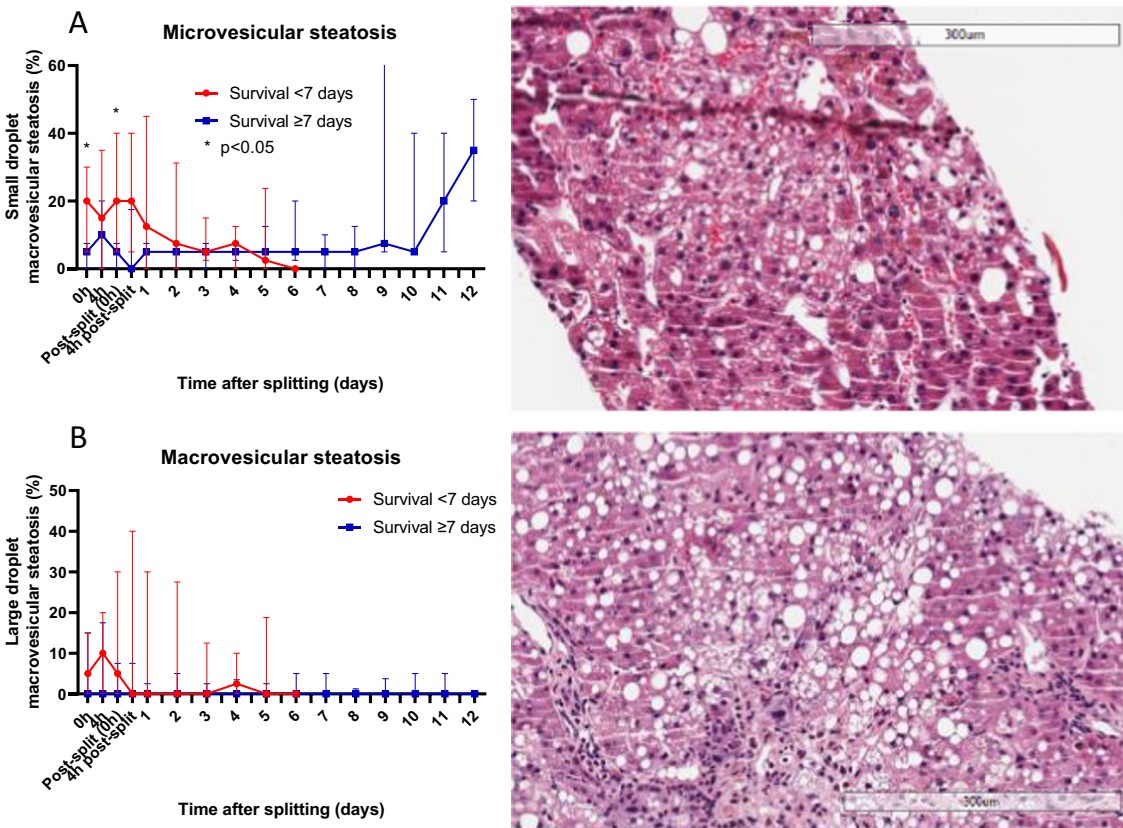

**Fig. 5 | Steatosis analysis of liver core biopsies taken throughout long-term perfusion of human split livers.** Slides were stained with haematoxylin and eosin and assessed by a blinded specialist pathologist for microvesicular (**A**) and macrovesicular (**B**) steatosis. Microvesicular steatosis seen on core biopsies taken before splitting was significantly less severe in livers that survived >7 days (median 5% [IQR 0–7.5%] vs 20% [IQR 5–35%], $p = 0.041$ at 0 h, Mann–Whitney U Test) (**A**). All grouped data are presented as median (IQR), $n = 20$ partial livers, 9 survived >7 days, 11 survived ≤7 days, *$p < 0.05$.

machine[4,14]. Using integrated automation and subnormothermic temperatures at 34 °C[6], 7 days of perfusion were achieved. By contrast, we utilised actual normothermic conditions (36 °C), which permits accurate simulation of physiological conditions and assessment without metabolic compromise. The perfusion of split livers as a model for research is also underutilised. Until now, the longest perfusion of matched partial livers was 6 h using an artificial oxygen carrier and a traditional ex situ split[15]. Our model could, in the future, be adapted to clinical transplantation with the advantage of a shorter cold ischaemia time with splitting occurring without interruption to vascular inflow and continuous perfusion of both livers in the long term, which allows adequate time for more sophisticated evaluation.

Long-term perfusion has given us an improved understanding of the changes that occur during ex situ machine perfusion under normothermic conditions. Despite nutritional and hormonal supplementation and filtration of the blood using dialysis, all livers in this study eventually failed. This suggests that there is a "Goldilocks" period within which these livers are at their best. During this period, the liver has been perfused long enough to allow adequate resuscitation and repair as well as assessment of the viability, but not too long such that liver quality declines and the irreversible deterioration towards organ failure has commenced. Utilising the viability criteria proposed by the VITTAL clinical trial in this study allowed us to identify the likely point of no return (typically around five days). Consideration must also be given to biliary viability, which is expected to be independent of hepatocellular viability[16]. As such, more sophisticated and less subjective criteria are required[3].

To this end, by grouping livers that survived >7 days and ≤7 days in this study, we were able to identify predictors of long-term survival using liver biochemistry, markers of synthetic liver function, liver haemodynamics, and histopathology. Organs that survived >7 days had significantly higher rates of bile production, higher levels of Factor-V, higher hepatic artery flows, and lower amounts of microvesicular steatosis. These changes were noticeable within the first 48–72 h of perfusion and represented potential targets for defining a signature for long-term survival. Not only does this have implications for the assessment of inherent organ quality, but this signature can be re-evaluated in real-time and guide us in the resuscitation and recovery of these livers in the long term.

The assessment of viability and long-term survivability of organs in this study is limited by a lack of validation by transplantation. Unfortunately, the livers available for research in this study were considered unsuitable for transplantation, and therefore evaluation by transplantation was not possible in this setting. However, this did afford us the opportunity to study these livers until failure, which has provided valuable information about the metabolic and haemodynamic patterns that occur towards organ demise. Importantly, achieving these results using marginal organs suggests that when we apply this to ideal organs that are suitable for transplantation, we are likely to be able to achieve more reliable and even longer survival results. This study also focussed primarily on organ viability from the perspective of hepatocellular function. Future work should perform a more detailed study of cholangiocellular function and evaluation of long-term biliary viability.

Finally, understanding why all these organs fail eventually is an area of ongoing research for our group. Towards organ failure, we have observed a characteristic pattern involving a rapidly rising lactate, increasing hepatic vascular resistance with low hepatic arterial and

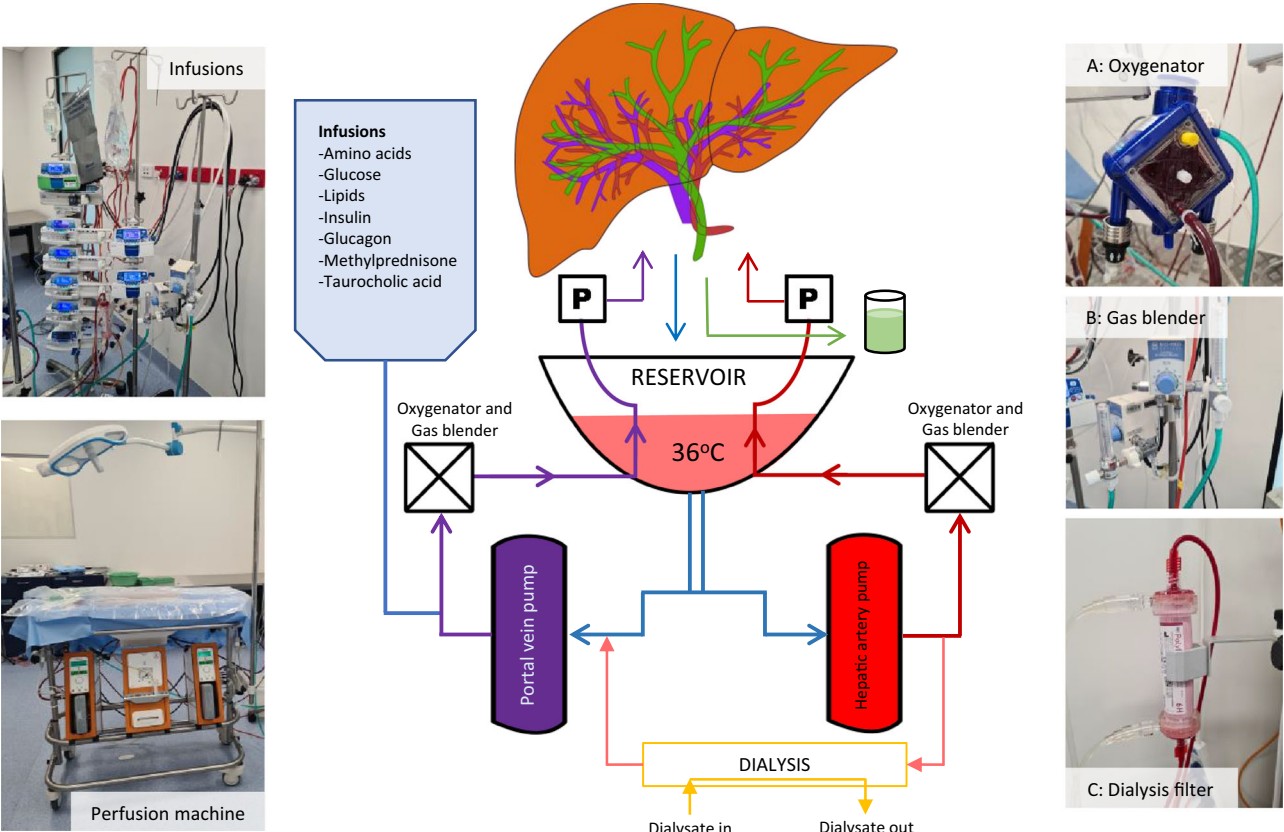

**Fig. 6 | Schematic diagram of our machine perfusion setup.** A commercially available organ perfusion system (Liver assist, Xvivo, Groningen, Netherlands) that uses a dual-pump system (**P**) and an open venous reservoir was modified for long-term perfusion by adding long-term oxygenators (**A**), a gas blender (**B**) and a dialysis filter (**C**).

portovenous flows, refractory perfusate acidosis, and unresponsive hypoglycaemia. Organ failure may be secondary to a missing key metabolic component required for maintenance energy homoeostasis or a deterioration in perfusate quality over time as blood products were not topped up or refreshed[17]. This model also remains non-physiological without a gut-liver axis or automated circadian variation[18]. These reasons may explain why most livers inevitably began failing after 5–7 days of perfusion. Alternately, there may be a missing growth factor stimulus that signals hepatocytes to regenerate[1]. Future work should aim to understand why these organs fail, unlock the potential to preserve these organs for weeks to months and provide more time for meaningful intervention and regeneration.

The ability to perfuse human split livers under normothermic conditions in the long term has enormous potential. In this study, we describe a functional model using a modified commercially available liver perfusion system to perfuse split human livers for >7 days. This model represents the longest-ever perfusion of human livers ex situ under normothermic conditions and has provided new information about how these organs can be evaluated for clinical use and why they fail in the long term. We describe a model suitable for ex situ perfusion of paediatric-sized organs and for expanding the applications of ex situ perfusion technology. Moreover, this technique has tremendous potential in the testing of therapeutics and paves the way for collaboration in the fields of transplantation, basic sciences and beyond.

## Methods
### Human livers
Human deceased donor livers that were declined for transplantation were accepted for research in accordance with an ethical protocol approved by the Sydney Local Health District Ethics Review Committee (X18-0523 & 2019/ETH08964). Consent for research was obtained at the time of consent for organ donation by the centralised donation organisation in Australia, DonateLife. We considered DBD and DCD livers and only declined an organ when there was clinical or biochemical evidence of cirrhosis. All livers were procured using our standard technique with aortic flushing using cold Soltran (Baxter Healthcare, Illinois, USA) and University of Wisconsin preservation solution (Belzer UW, Bridge to Life, Columbia, USA), followed by static cold storage for transfer to our centre. No organs from executed prisoners were used.

### Modified perfusion machine
We modified a commercial liver perfusion system (Liver assist, Xvivo, Groningen, Netherlands) as previously described using commercially available equipment[9–11]. This system utilises a dual-pump configuration with an open venous reservoir and thermoregulatory unit. The modifications are summarised in Fig. 6. We replaced the commercially-provided oxygenators with long-term oxygenators (Quadrox-iD Pediatric, Macquet, Getinge Group, Rastatt, Germany) and added a gas blender (Low Flow 2003 Series, Bio-Med Devices, Connecticut, USA) and paediatric flow regulators for precision mixing of compressed air with oxygen. We also connected a dialysis filter in parallel with adjustable input and output controls (Prismaflex or Polyflux, Baxter Healthcare, Illinois, USA).

### Perfusion protocol
A red blood cell-based perfusate was prepared using 4 units of donated packed red blood cells, 2 units of donated fresh frozen plasma, 200 ml of 20% albumin and 1 L of normal saline. The system was anticoagulated throughout using enoxaparin (100 mg twice daily, Clexane,

Sanofi-Aventis, Auckland, NZ), and the perfusate was "cleaned" prior to connecting the liver by continuous circulation of perfusate through the dialysis filter and correction of electrolyte abnormalities. Once physiological conditions were achieved, the donated human livers were flushed with normal saline before immediate perfusion by portal vein and hepatic artery cannulas. Utilising a pressure-controlled system, targets of 60 mmHg and 8 mmHg were chosen for the hepatic artery and portal vein, respectively. An 18-20F arterial cannula (Organ Assist, Groningen, Netherlands) was used for the hepatic artery and a 25F portal vein cannula (Organ Assist, Groningen, Netherlands) for the portal vein. This aimed to achieve a hepatic artery flow of >400 ml/min and a portal vein flow of >1.2 L/min. Controlled rewarming was performed with a 1° increase in temperature per hour for 4 h (from the initial 32 °C to 36 °C) to maintain perfusion in a temperature range conducive to red blood cell survival and minimise the effects of ischaemia reperfusion injury[12,19].

Antibiotic prophylaxis was provided with cefazolin (1 g daily, AFP Pharmaceuticals, NSW, AUS) to minimise bacterial contamination. Taurocholic acid (7.7 mg/h, Cayman Chemical Company, Michigan, USA) and methylprednisolone (Solu-Medrol reconstituted to 10 mg/ml, 21 mg/h, Pfizer, NSW, AUS) were added to promote survival. Nutritional support was provided using continuous infusions of amino acids (Synthamin 17, Baxter Healthcare, Illinois, USA) and lipids (Clinoleic 20%, Baxter Healthcare, Illinois, USA). A multivitamin (Cernevit, 1 vial, Baxter Healthcare, Illinois, USA) and thiamine (Thiamine Hydrochloride, 100 mg/ml, 1 vial, Biological Therapies, VIC, AUS) were administered once daily. Lactate, pH, glucose, pO2 and pCO2 were measured using a blood gas analyser to facilitate real-time modification of infusions (RAPIDPoint 500, Siemens Healthengineers, Norwood, Massachusetts, USA). Glucose levels were manually maintained in the range of 5–15 mmol/L using titratable infusions of insulin (Actrapid 100IU/ml diluted to 2IU/ml, typical range: 2–6 IU/h, Novo Nordisk, Auckland, NZ), glucagon (GlucaGen reconstituted to 1 mg/ml and diluted to 20 µg/mL, typical range: 40–120 µg/h, Novo Nordisk, Auckland, NZ) and glucose (10%, typical range 5–20 ml/h, Baxter Healthcare, Illinois, USA). Acid-base balance and oxygenation were maintained manually by adjustment of oxygen-compressed air ratios and regulation of gas flow, aiming for a pH of 7.3–7.45 and a pO2 of 100–200 mmHg. Dialysis filtration using dialysate without supplementation (Hemosol B0, Baxter Healthcare, Illinois, USA) was manually adjusted to maintain potassium in the range of 3–5 mmol/L and haemoglobin in the range of 55–65 g/L. All manual adjustments were protocolised and algorithmic where possible (Supplementary Fig. 5). Red blood cell top-ups were not required. Severe acidosis (typically at the start or end of perfusion was treated with 10 ml aliquots of sodium bicarbonate 8.4% (8.4 g in 100 ml, Phebra, NSW, AUS). Arterial vasospasm (relating to the handling of the liver or administration of medications) was treated with 2.5 mg boluses of verapamil (Isoptin, 5 mg/2 ml, Viatris, NSW, AUS).

### Liver splitting and perfusion of partial livers

For livers that were split, splitting was performed the day after the commencement of perfusion, which corresponded to between 12 and 16 h of whole liver perfusion. The liver was split ex situ with continuous dual perfusion using the technique we have previously described[9–11]. In brief, the liver was divided into a left lateral segment graft (LLSG, segments 2 and 3) and an extended right graft (ERG, segments 1 and 4–8). The coeliac trunk was allocated to the LLSG, and the main portal vein was kept with the ERG. A 10-12F Foley catheter was used to cannulate the right hepatic artery (for the ERG), and an 18F Foley catheter was used to cannulate the left portal vein (for the LLSG). The common bile duct (with the ERG) and the left hepatic duct (with the LLSG) were cannulated for the collection of bile.

After completion of parenchymal and vascular dissection, the ERG was transferred to a second modified liver perfusion system. Nutritional support was adjusted for the size of the organs. As for the whole liver, pressure-controlled targets were set at 60 mmHg and 8 mmHg for the hepatic artery and portal vein, respectively. This was manually adjusted to achieve minimum arterial flows of 150 ml/min and portovenous flows of 300 ml/min for the LLSG and arterial flows of 150 ml/min and portovenous flows of 800 ml/min for the ERG.

All livers were continuously assessed using the viability criteria proposed in the VITTAL clinical trial (lactate ≤2.5 mmol/L, and two or more of: bile production, pH ≥ 7.30, glucose metabolism, hepatic arterial flow ≥150 ml/min and portal vein flow ≥500 ml/min, or homogeneous perfusion)[2]. Perfusion was continued until the organs were clearly non-viable, with an exploratory focus on understanding the physiological changes towards organ demise. Perfusion was ceased when perfusate lactate was >10 mmol/L or exponentially rising, and there was a cessation of bile production and unresponsive hypoglycaemia. Liver viability according to the DHOPE-COR-NMP trial (lactate <1.7 mmol/L, pH 7.35–7.45, bile production >10 ml and bile pH >7.45) was also assessed during perfusion to include an evaluation of biliary viability[12]. Our long-term perfusion protocol for split human livers is summarised in Fig. 7.

### Assessment of liver function

Perfusate samples and liver core biopsies were collected daily for assessment of biochemical markers and changes over time. All tests, unless specified, were measured by the clinical laboratory of the Royal Prince Alfred Hospital, Sydney, Australia. Liver function tests (bilirubin, ALT, GGT, ALP) were assessed for evidence of liver injury over time. Liver synthetic function was assessed using perfusate lactate, PT and levels of Factor-V. Bile production was measured every 4–6 h and adjusted to the weight of each partial liver to be expressed as ml/h/kg. Perfusate and bile pH and glucose were assessed using a blood gas analyser (RAPIDPoint 500, Siemens Healthengineers, Norwood, Massachusetts, USA).

Oxygen consumption was estimated using Henry's Law, with individual calculations of free and bound oxygen content in the hepatic artery, portal vein and hepatic vein: $[O_2 = pO_2 \times k + sO_2 \times Hb \times c]$ where $pO_2$ and $sO_2$ are the partial pressure of oxygen and oxygen saturations, respectively, and $k$ is the oxygen solubility coefficient, and $c$ is the oxygen-carrying capacity of haemoglobin. This was utilised to calculate the overall oxygen extraction/consumption rate using the hepatic artery, portal venous and hepatic vein flow and adjusted to the weight of the livers: [20,21]

$$[Oxygen\ Consumption = Flow_{HA} \times O2_{HA} + Flow_{PV} \times O2_{PV} - Flow_{HV} \times O2_{HV}].$$

Hepatic tissue ATP and glycogen content were determined using the ATP Bioluminescent Kit (FLAA Sigma-Aldrich) and Glycogen Assay Kit (MAK016, Sigma-Aldrich), respectively, according to the manufacturer's instructions (Supplementary Notes 1).

### Histology

Biopsies were fixed in formalin, embedded in paraffin, sectioned at 4 µm and stained with haematoxylin and eosin (H&E) and periodic acid-Schiff (PAS) stains. Histological assessment was performed by a blinded specialist pathologist with expertise in liver transplantation and liver disease, and any queries were resolved by conferral with a second specialist pathologist. Each slide was assessed and scored for macrovesicular and microvesicular steatosis, coagulative necrosis, hepatocyte detachment and glycogen depletion using a structured quantitative scoring system[22]. Ki67 immunohistochemistry was performed on selected slides to assess for evidence of cellular proliferation. Bile duct biopsies were examined for mural stromal necrosis and biliary epithelial injury.

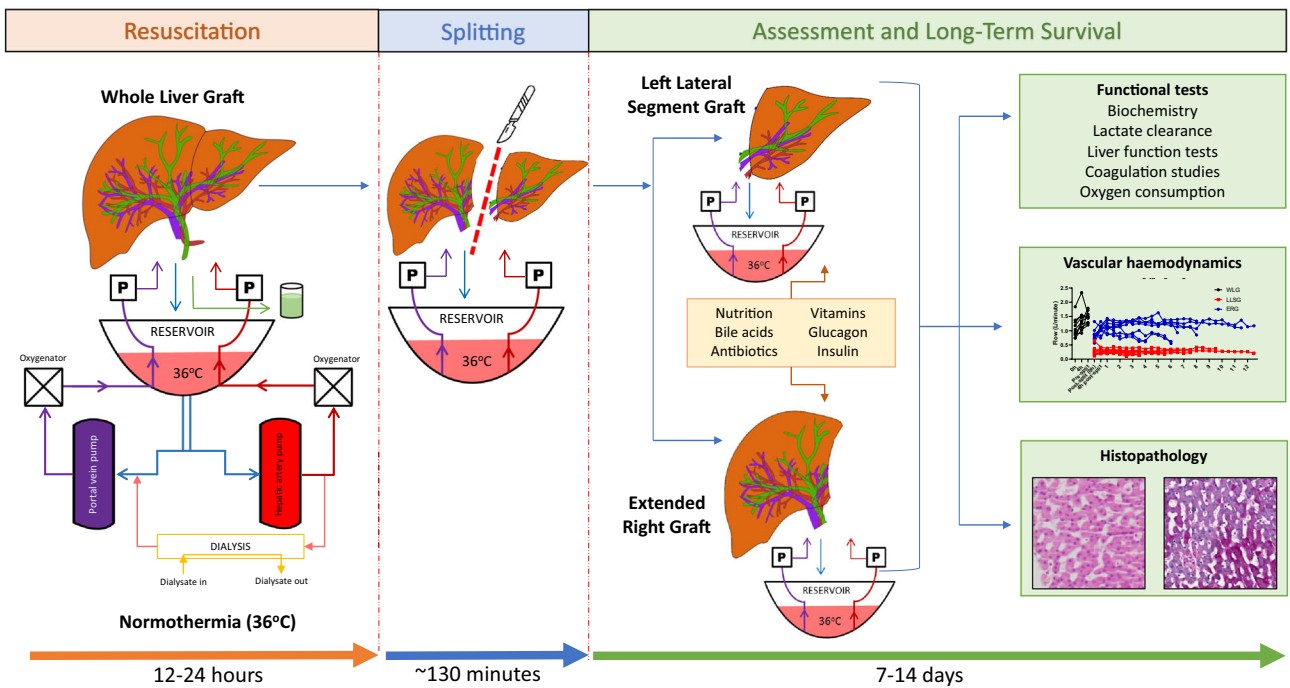

**Fig. 7 | Summary of experimental design for the long-term perfusion of human split livers.** Donated human livers are resuscitated under normothermic conditions for 12–24 h before a conventional split is performed. The left lateral segment graft and extended right graft are then perfused on separate perfusion machines for long-term assessment of liver function using functional tests, haemodynamic evaluation and histopathology.

## Statistics

Graphs were created and statistical analysis was performed using GraphPad Prism 9 (GraphPad Software, San Diego, California, USA). Data for each partial liver were grouped according to time after splitting and expressed as mean ± standard deviation or median (IQR) as appropriate. Continuous variables were compared at each grouped timepoint using an unpaired two-sided $t$-test or a Mann–Whitney U Test as appropriate. Results were considered significant if $p < 0.05$.

## Reporting summary

Further information on research design is available in the Nature Portfolio Reporting Summary linked to this article.

## Data availability

All data needed to reproduce this study can be found in the manuscript, figures, and supplementary information. The source data for Tables and Figures in this study are provided with this paper in the Source Data File. Source data are provided with this paper.

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

## Acknowledgements

We gratefully acknowledge the support and assistance provided by M. Xiang, N. Koutalistras and L. Gao. We would also like to thank DonateLife and all organ donors and their families for their support, without which this research would not be possible. Financial support was provided by the Royal Prince Alfred Hospital Transplant Institute. Some equipment used in this study was provided by Johnson and Johnson (Harmonic Scalpel, Eschelon Stapler). N.L. is supported by the Australian Government Research Training Program Stipend Scholarship. M.L. is supported by the University of Sydney University Postgraduate Award. M.C.-C. is supported by the NSW Health Early-Mid Career Research Grant.

## Author contributions

N.L. participated in the research design, performed the research, analysed the data and wrote the paper. M.L., C.D. performed the research, analysed the data and reviewed the manuscript. A.J., M.C.-C., S.T., J.H., N.M., P.Y., S.C., C.W. performed the research and reviewed the manuscript. L.L., K.L., J.K., G.M. and M.C. participated in the research design and critically reviewed the manuscript. C.P. participated in research design, performed the research, analysed the data and reviewed the manuscript.

## Competing interests

The authors declare no competing interests.
