## [Peer Review File · Nature Communications]

Long-term ex-situ normothermic perfusion of human split
livers for more than 1 weekREVIEWER COMMENTS

Reviewer #1 (Remarks to the Author):

First of all, I would like to congratulate the authors on the completion of this experimental study about long-term normothermic machine perfusion (NMP) with human split livers. These long perfusions must have been a tremendous amount of work and great teamwork. Please, find below my comments on the manuscript, separated into major and minor comments.

Major comments

1) In general, the authors claim to present “a reliable long-term NMP model for split livers”, but only 11/20 were viable after 5 days. This does not indicate that it is a very reliable and functional model. The model is definitely functional to a certain extent, but they should emphasize a bit more that it is still a model in development. To model should also have been used with a few whole livers to determine the effect of the model without an extra intervention (the splitting).

Abstract:

2) In the abstract is stated that it is a model for long-term ex-vivo perfusion for split livers, but only slightly more than half of the split livers (11/20) were still viable after 5 days. This makes me wonder if the model is really that functional for long-term perfusion. Maybe, this model should first be performed with whole livers to decide how functional the model is, before doing an intervention (see the main comment).

3) The median ex-vivo survival time is given in the abstract, but it might be more useful to show the median ex-vivo viability time, because the perfusion was only extended after viability failed for extra information about the understanding of the physiological changes towards graft demise.

4) Moreover, the term “survival” suggest that this is model of dying livers with a (still) limited capacity/duration to preserve organs ex situ.

5) The grafts that survived >7 days showed higher bile production, but this was a criterion for survival, so it cannot also be the outcome.

Introduction:

6) As stated in comments 1 and 2, with only 11/20 surviving ≥ 5 days of NMP, how reliable is this model really? The authors should down tune their interpretation of their findings.

7) When reading the manuscript, the main objective of this project remains unclear? Was it the aim to develop a device for long-term clinical machine perfusion or was it the aim to develop a research model with machine perfusion of comparable liver parts? These are two different objectives that have different requirements.

Methods:

8) The perfusion protocol is quite vague, while it was advocated to be a reproducible protocol. For example:

a. The perfusion pressures are stated in sentence 135, but the size of the cannulas is not mentioned, while the combination of pressure and cannula size is important.

b. Verapamil and bicarbonate are administered during perfusion according to Figure 1, but they are not mentioned in the method section.

c. The infusion doses of taurocholic acid and methylprednisolone are mentioned, but not from nutritional supplementation, glucose, insulin and glucagon. These last ones are depended on the liver size and perfusate glucose values, respectively, however more details would make it a more comparable and reproducible model.

d. Antibiotics were added to promote survival (as it is stated now; P7, L140-141) or to avoid bacterial contamination/growth/infections?

e. Dialysis filtration was mentioned, but not if there was any suppletion of dialysate.

9) Why was chosen for the VITTAL clinical trial criteria? These focus on short-term NMP and only focus on hepatocyte function. Other viability criteria, including cholangiocellular viability criteria, are used nowadays and might give a better insight into the function and viability of the split livers.

10) Biopsies were only taken from the liver parenchyma and not from the bile ducts? (i.e. at the start and end of NMP). Assessment bile ducts (canaliculi) in parenchymal biopsies is not representative for morphology of the larger bile ducts and this should be acknowledged. If biopsies of the larger bile ducts were taken, the results of the histological analysis should be added to the manuscript.

11) Statistical p-value was based on-sided or two-sided testing? Please specify.

Results:

12) Bile production is described as significantly different between graft survival >7 days or <7 days, however, this was also a criterion for graft survival, and the significance was only seen on 3 time points. Because it was a criterion for graft survival, you cannot also use it to show a significant outcome between the two groups, because you selected the grafts on this.

13) Perfusate glucose is not significantly different between 7 days or <7 days, however, this is supplemented during perfusion to keep in range of 5-15 mmol/l. If you want to say something about this, you should compare the amount of glucose administered. The same for the perfusate pH. If bicarbonate is administered during perfusion to keep the pH between 7.3-7.45, the amount of bicarbonate is more important than if there is a difference in pH.

Discussion:

14) In the first sentence of the Discussion, the authors state that they have presented “a reliable and reproducible model of long-term normothermic machine perfusion”. However, only 9 of the 20 grafts survived >7 days and sometimes one part of the split liver survived while the other part of the same liver did not. This is not consistent with “a reproducible model”. Please, rephrase.

15) Second sentence of the Discussion: “Using livers that were not usable at our centre for transplantation, ex-vivo preservation was achieved routinely for longer than seven days and up to a maximum of 13 days.”. This conclusion/statement is not supported by the results. Only 6 of the 20 liver grafts were viable up to 7 days, which is not the same as “routinely longer than 7 days”. Moreover, only one of the 20 grafts was viable up to 13 days

16) In relation to comment #13, how reliable is this model of simultaneous perfusion of two partial grafts to study repair and regeneration prior to transplantation if both parts do not have similar survival times?

17) In the discussion part it is mentioned that this long-term perfusion protocol can be used for meaningful/sophisticated evaluation of livers. It is unclear to me what is meant by this. It assumes that short-term NMP is not good enough for viability assessment. This should be described in a different way.

18) It is mentioned that there is a “Goldilocks” period for these livers, but again, this could be caused by the model. It should be compared with whole liver perfusion to find out if the intervention was part of this maximum period or not.

Minor Comments:

19) Red cells should be red blood cells.

20) Abbreviations (page 2): “alanine” is missing after “ALT:...”. GGT: “tranferase” is missing

21) There is consensus in the field that we should not use the term “ex vivo” (Latin for “outside the living”) when talking about organs from deceased donors. The preferred term is “ex situ” (Karangwa, et al. Am J Transplant 2016)

22) Method section: Coagulation studies INR/PT were mentioned as a test for liver synthetic function but were not provided in the results.

23) Method section: Please provide a reference of an article that describes haemolysis due to rewarming. What is the evidence that the temperature should be increased with 1 degree Celsius per hour? Why start at 32 degrees and not 20 or directly at 36?

24) Method section: Oxygen consumption was estimated using Henry’s law; shouldn’t this be in combination with Fick’s law? Otherwise, it is calculated only on dissolved oxygen. Also, the formula is not complete, because it also takes into account the blood flows through the liver.

25) P8, L179: “Liver and bile pH...” should be “Perfusate and bile pH...”

26) Result section: AST is mentioned in the method section, but not shown in the figures.

27) Figure 2: Infusion of vitamins is not mentioned in the manuscript.

28) The authors describe a decrease in HA flow with the same HA pressure, however, this is not visible in the graph (Figure 4). It looks more like the opposite. If this is not the case, maybe the graphs are not that clear.

Reviewer #2 (Remarks to the Author):

The manuscript “Long-term ex-vivo normothermic perfusion of human split livers: a unique model to study new therapeutics and increase the number of available organs” by Dr Lau and colleagues reports development of a novel protocol and experience with a week-lasting

discarded donor liver perfusions.

The authors describe series of 10 discarded donor livers, that underwent splitting during the NMP, with subsequent commencement of NMP using another device. This approach resulted in 20 extended perfusions. The manuscript is well written and presents a novel data that expanding on the work authors' previously publications.

The extended NMP model offers tremendous potential to study resuscitation and regeneration of suboptimal donor livers. The split-organ approach in addition allows researchers to design NMP studies to test novel interventions on livers that has got an identical risk profile. Dr Lau and colleagues should be congratulated for such outstanding contribution and pushing boundaries of the liver perfusion. The manuscript represents an abundance of hard work, and the results are impressive, albeit understated.

This paper makes one of the first to attempt to describe features typical of liver failure during NMP. The features described are universal but could be argued as very end-stage signs of liver failure, namely lactate >10 mmol/L and hypoglycaemia refractory to intervention. This might impact the results by categorizing organs as viable whilst their function is already progressively deteriorating.

The authors present some features of livers able to survive 7 days, but there is a lack of discussion in why a liver fails during NMP and what separates the organs that thrive. Some organs were declined based on donor age and the NMP was commenced following a short period of cold ischaemia only (<5 hours). Could authors speculate what might be the reasons such livers' perfusion did not exceed 168 hours? Less than 50% of the organs were able to tolerate perfusion beyond 7 days, and the overall median perfusion time was 5 days. The main shortcoming of the paper is omission of important details that would allow readers to replicate the experiments. The aims and methodology are well defined and described, with a clear objective to explore and define the features of organs capable of surviving >7 days during NMP. However, if a reader wished to replicate the results described, there is a lack of detail in the perfusion protocol to facilitate this. Key missing aspects include formulations of drugs (& brands), flowcharts for maintaining the desired perfusate parameter levels (i.e. glucose, and haemoglobin) and these and other troubleshooting issues. As such, the statement that the presented extended NMP protocol for split human livers is both reproducible and reliable is not supported by the data

included.

Authors did not provide details about the oxygen delivery and consumption during the perfusions. Were there any changes in the perfusate quality and oxygen carrying capacity throughout the course of the perfusions? Were all perfusions performed without the need to top up the red blood cells, or replacement of blood products? What were the levels of methaemoglobin or carboxyhaemoglobin? These aspects would be worth highlighting in the Discussion.

An important result described is the difference in hepatic arterial flow between livers that survived 7 days versus organs that didn't. The authors describe livers that were more likely to survive as having increased arterial flow. From a purely physiological perspective, hepatic arterial flow is weight dependent (about 0.25 ml/g /min). Could authors explore how the differences in flow related to liver weight or whether suitable intervention to increase flows may improve the likelihood of salvaging the failing livers? Presenting the vascular flows in relation to the liver mass (similarly to bile production) might reveal some correlation not apparent at the current data presentation form.

The Discussion should mention which perfusion and perfusate parameters were controlled and adjusted, or which were set and not changed (hepatic artery and portal vein pressures), or which are normalised by dialysis filter (pH). This would make clearer why some graphs in the Figure 6 appear to be similar in viable and failing livers.

In summary, the maintenance of 10-day normothermic liver perfusion is a tremendous success not yet achieved / published by others (and the 328 hours is the world-longest NMP liver perfusion). However, the presented data lacks detail, which limits the experiments reproducibility and manuscript informative value. The authors should provide more details to help readers to appreciate the complexity and labour-intensity of the experiments and allow to replicate them. The Discussion should comment on some aspect of extended perfusions that others might struggle to overcome. These changes would help guide others and are likely to increase the impact of this important paper.

Minor comments

The Figure 3 shows several closely packed graphs which do not allow to inspect the data in detail

The term *ex situ* would be preferred to *ex vivo* (Karangwa et al 2016)

DCD is a widely used abbreviation for donation after circulatory death, so would be preferred to DCDD.

If the liver is not transplanted, then it is not a graft and would be better referred to as liver / organ.

Point-by-point response to reviewer comments

We would like to thank the reviewers of *Nature Communications* for their excellent comments and feel that the paper is now stronger as a result. We have addressed all the questions raised to the best of our abilities below.

Reviewer #1 (Remarks to the Author):

First of all, I would like to congratulate the authors on the completion of this experimental study about long-term normothermic machine perfusion (NMP) with human split livers. These long perfusions must have been a tremendous amount of work and great teamwork. Please, find below my comments on the manuscript, separated into major and minor comments.

Major comments

1) In general, the authors claim to present “a reliable long-term NMP model for split livers”, but only 11/20 were viable after 5 days. This does not indicate that it is a very reliable and functional model. The model is definitely functional to a certain extent, but they should emphasize a bit more that it is still a model in development. To model should also have been used with a few whole livers to determine the effect of the model without an extra intervention (the splitting).

We thank Reviewer #1 for their comments and feedback.

We recognise that our model is not entirely mature and therefore not a truly reliable or reproducible model. The language has therefore been softened to put forward the model as ‘functional’ rather than ‘reproducible’ in the abstract (page 3), introduction (page 5) and discussion/conclusion (page 12 and page 15).

The use of the model with whole livers has been added (page 6-7) demonstrating that 3/3 livers survived >7 days. This will be discussed in response to further comments below.

Abstract:

2) In the abstract is stated that it is a model for long-term ex-vivo perfusion for split livers, but only slightly more than half of the split livers (11/20) were still viable after 5 days. This makes me wonder if the model is really that functional for long-term perfusion. Maybe, this model should first be performed with whole livers to decide how functional the model is, before doing an intervention (see the main comment).

We recognise that our model is not entirely mature and therefore should not be considered a ‘reliable or reproducible’ model. This language has been toned down throughout the text as discussed above.

The use of our long-term protocol for whole livers has been added to the manuscript to demonstrate functionality of the model without splitting. We demonstrated long-term survival (>7 days) for 3 whole livers evidenced by lactate clearance and bile production. These data have been added to the results section (page 6-7) and a figure added to demonstrate lactate clearance and overall survival (Supplementary Figure 1).

3) The median ex-vivo survival time is given in the abstract, but it might be more useful to show the median ex-vivo viability time, because the perfusion was only extended after viability failed for extra information about the understanding of the physiological changes towards graft demise.

The median ex-situ viability time has been added to the abstract as requested (125 hours) (page 3).

4) Moreover, the term “survival” suggest that this is model of dying livers with a (still) limited capacity/duration to preserve organs ex situ.

We agree that as it stands, this continues to be a model of ‘dying livers’ with a limited capacity. There is no doubt that all livers in this study eventually failed. This is an ongoing area of active research for our group to determine whether there is a missing metabolic component for energy homeostasis or perhaps a missing growth factor stimulant. This has been expanded in the discussion (page 14-15).

5) The grafts that survived >7 days showed higher bile production, but this was a criterion for survival, so it cannot also be the outcome.

The presence of bile production is one of the secondary criteria for preserved viability in the VITTAL clinical trial, and was used in this study to assess viability. However, the classification of viability and survival did not rely on the presence or absence of bile production in any cases. Supplementary Table 3 has been added to outline the details of viability and survivability for all livers to clarify this. Ultimately, lactate was the determining factor for long-term hepatocellular viability in all cases. Indeed, these criteria were also not designed for continuous use in the long-term, and did not distinguish between organs that had low or high rates of bile production. In this study, the rate of bile production in ml/h/kg liver was significantly higher in the days following splitting for the grafts that survived >7days. We believe this represents a marker of superior organ quality and identifies those livers that are thriving the ex-situ environment. This has been clarified in the manuscript on page 7.

Introduction:

6) As stated in comments 1 and 2, with only 11/20 surviving ≥ 5 days of NMP, how reliable is this model really? The authors should down tune their interpretation of their findings.

As discussed above, we recognise that our model is not entirely mature and therefore should not be considered a ‘reliable or reproducible’ model. This language has been toned down in the abstract (page 3), introduction (page 5) and discussion/conclusion (page 12 and 15).

7) When reading the manuscript, the main objective of this project remains unclear? Was it the aim to develop a device for long-term clinical machine perfusion or was it the aim to develop a research model with machine perfusion of comparable liver parts? These are two different objectives that have different requirements.

In this study, our aim was to develop a model of long-term normothermic ex-situ perfusion of split livers. We see this as a research model at this stage but with important implications for the development of clinical long-term perfusion. This has been clarified in the introduction (page 5) and discussion/conclusion (page 15).

Methods:

8) The perfusion protocol is quite vague, while it was advocated to be a reproducible protocol. For example:

a. The perfusion pressures are stated in sentence 135, but the size of the cannulas is not mentioned, while the combination of pressure and cannula size is important.

The technical details for our protocol have overall been enhanced such that a reader could replicate our model.

We used an 18-20F arterial cannula (Organ Assist, Gronigen, Netherlands) for the hepatic artery, and a 25F portal vein cannula (Organ Assist, Gronigen, Netherlands) for the portal vein. After splitting, we used a 10-12F Foley catheter to cannulate the right hepatic artery (in the ERG) and an 18F Foley catheter to cannulate the left portal vein (for the LLSG).

These details have been added to the methods section (page 17 and 18).

b. Verapamil and bicarbonate are administered during perfusion according to Figure 1, but they are not mentioned in the method section

Severe acidosis (typically at the start or end of perfusion) was treated with 10ml aliquots of sodium bicarbonate 8.4%. Arterial vasospasm (relating to handling of the liver or administration of medications) was treated with 2.5mg boluses of a calcium channel blocker (verapamil hydrochloride 5mg/2ml). These details have been added to the methods section (page 18). These were not routine continuous infusions and therefore have also been removed from Figure 1 (now Figure 6)

c. The infusion doses of taurocholic acid and methylprednisolone are mentioned, but not from nutritional supplementation, glucose, insulin and glucagon. These last ones are dependent on the liver size and perfusate glucose values, respectively, however more details would make it a more comparable and reproducible model.

Glucose, insulin and glucagon were manually titrated to maintain a glucose level of 5-15mmol/L. Typical ranges have been added to the methods section to increase reproducibility (Insulin (Actrapid 2IU/ml) typical range: 2-6IU/h), glucagon (20µg/mL, typical range: 40-120µg/h) and glucose (10%, typical range 5-20ml/h) (page 18). The typical algorithm utilised for manual adjustment of these parameters has been added (referenced in text page 18, Supplementary Figure 5).

d. Antibiotics were added to promote survival (as it is stated now; P7, L140-141) or to avoid bacterial contamination/growth/infections?

Antibiotic prophylaxis was provided with cephazolin (1g daily) to minimise bacterial contamination. This has been clarified in the methods section (page 17)

e. Dialysis filtration was mentioned, but not if there was any supplementation of dialysate.

Dialysate was not supplemented. This has been clarified in the manuscript, and details of dialysate fluid provided (Hemosol B0, Baxter, Deerfield, Illinois, USA) (page 18)

9) Why was chosen for the VITTAL clinical trial criteria? These focus on short-term NMP and only focus on hepatocyte function. Other viability criteria, including cholangiocellular viability criteria, are used nowadays and might give a better insight into the function and viability of the split livers.

The VITTAL clinical trial criteria were chosen as they were easily reproducible throughout perfusion (although only designed for short-term use) and validated by a clinical trial. We recognise that these criteria focus on hepatocellular function. We have added evaluation of livers using a hepatobiliary criteria (DHOPE-COR-NMP trial¹) to include an assessment of cholangiocellular viability (Results: Page 7, Methods: page 19, Supplementary Table 3). Notably, this includes bile pH as a viability criterion.

We found that all livers apart from the 2 that failed due to a technical error also met these criteria and produced bile with a pH >7.40 (Results: page 7, Supplementary Table 3)

¹ van Leeuwen, O. B. *et al.* Transplantation of High-risk Donor Livers After Ex Situ Resuscitation and Assessment Using Combined Hypo- and Normothermic Machine Perfusion: A Prospective Clinical Trial. *Ann Surg* **270**, 906-914, (2019).

10) Biopsies were only taken from the liver parenchyma and not from the bile ducts? (i.e. at the start and end of NMP). Assessment bile ducts (canaliculi) in parenchymal biopsies is not representative for morphology of the larger bile ducts and this should be acknowledged. If biopsies of the larger bile ducts were taken, the results of the histological analysis should be added to the manuscript.

The focus of this manuscript was primarily hepatocellular viability and survival however we recognise the importance of biliary assessment. Biopsies from large bile ducts have been added. We found that biliary injury was present in bile duct biopsies with evidence of mural stromal necrosis, however 80% of livers demonstrated intact biliary epithelium suggesting preserved biliary tree integrity (Methods: page 20, Results, page 10, Supplementary Figure 2).

11) Statistical p-value was based on-sided or two-sided testing? Please specify.

Statistical p-values were based on two-sided testing for normally distributed data. This has been clarified in the methods section (page 21)

Results:

12) Bile production is described as significantly different between graft survival >7 days or <7 days, however, this was also a criterion for graft survival, and the significance was only seen on 3 time points. Because it was a criterion for graft survival, you cannot also use it to show a significant outcome between the two groups, because you selected the grafts on this.

The presence or absence of bile production is one of the secondary criteria for preserved viability in the VITTAL clinical trial, and was used in this study to assess viability. However, the classification of viability and the length of survival did not rely on the presence or absence of bile production in any cases, but rather seemed to depend on perfusate lactate levels and acidosis or hypoglycaemia. Supplementary Table 3 has been added to outline the details of viability and survivability for all livers to clarify this.

In this study, the rate of bile production in ml/h/kg liver was significantly higher at 3 time points in the days following splitting for the grafts that survived >7days. We believe this represents a marker of superior organ quality and identifies those livers that are thriving in the ex-situ environment. This has been clarified in the manuscript on page 7.

13) Perfusate glucose is not significantly different between 7 days or <7 days, however, this is supplemented during perfusion to keep in range of 5-15 mmol/l. If you want to say something about this, you should compare the amount of glucose administered. The same for the perfusate pH. If bicarbonate is administered during perfusion to keep the pH between 7.3-7.45, the amount of bicarbonate is more important than if there is a difference in pH.

The levels of perfusate glucose and pH were manually maintained throughout perfusion which explains why there are no differences between the two groups. These were non-contributory to the results and therefore have been removed (Results page 11, Figure 4, Supplementary Figure 2).

Discussion:

14) In the first sentence of the Discussion, the authors state that they have presented “a reliable and reproducible model of long-term normothermic machine perfusion”. However, only 9 of the 20 grafts survived >7 days and sometimes one part of the split liver survived while the other part of the same liver did not. This is not consistent with “a reproducible model”. Please, rephrase.

As discussed above, we recognise that our model is not entirely mature and therefore should not be considered a ‘reliable or reproducible’ model. This language has been toned down in the discussion/conclusion as requested (page 12).

15) Second sentence of the Discussion: “Using livers that were not usable at our centre for transplantation, ex-vivo preservation was achieved routinely for longer than seven days and up to a maximum of 13 days.”. This conclusion/statement is not supported by the results. Only 6 of the 20 liver grafts were viable up to 7 days, which is not the same as “routinely longer than 7 days”. Moreover, only one of the 20 grafts was viable up to 13 days

As discussed above, we recognise that our model is not entirely mature and therefore should not be considered a ‘reliable or reproducible’ model. This language has been toned down in the discussion/conclusion as requested (page 12).

16) In relation to comment #13, how reliable is this model of simultaneous perfusion of two partial grafts to study repair and regeneration prior to transplantation if both parts do not have similar survival times?

We recognise that there is much we do not understand about long-term perfusion and the factors that contribute to long-term survival. We believe differences in survival of the two partial grafts are related to either a missing metabolic component for energy homeostasis or a missing growth factor stimulant. It is possible that the smaller left lateral segment graft has less energy stores, or that the extended right graft has greater build up of a metabolic toxin. Overall, we agree that our model is not entirely mature and therefore should not be considered a ‘reliable or reproducible’ model. This has been toned down throughout the text (abstract (page 3), introduction (page 5) and discussion/conclusion (page 12 and 15)) and this concept discussed further in the discussion section (page 15).

17) In the discussion part it is mentioned that this long-term perfusion protocol can be used for meaningful/sophisticated evaluation of livers. It is unclear to me what is meant by this. It assumes that short-term NMP is not good enough for viability assessment. This should be described in a different way.

We believe that long-term perfusion may facilitate more sophisticated viability assessment, repair of organs prior to transplantation, and build on the work already done in the field of short-term viability testing. This has been rephrased and clarified as requested on page 12.

18) It is mentioned that there is a “Goldilocks” period for these livers, but again, this could be caused by the model. It should be compared with whole liver perfusion to find out if the intervention was part of this maximum period or not.

As discussed above, the use of our long-term protocol for whole livers has been added to the manuscript to demonstrate functionality of the model without splitting. We demonstrated long-term survival (>7 days) for 3 whole livers evidenced by lactate clearance and bile production. These data have been added to the results section (page 6-7) and a figure added to demonstrate lactate clearance and overall survival (Supplementary Figure 1). By comparison to these data, the ‘goldilocks’ period continues to hold true, with a likely point of no return (typically around five days) for all organs partial or whole.

Minor Comments:

19) Red cells should be red blood cells.

This has been corrected throughout the text.

20) Abbreviations (page 2): “alanine” is missing after “ALT:...”. GGT: “tranferase” is missing

This has been corrected on page 2.

21) There is consensus in the field that we should not use the term “ex vivo” (Latin for “outside the living”) when talking about organs from deceased donors. The preferred term is “ex situ” (Karangwa, et al. Am J Transplant 2016)

This has been corrected throughout the text.

22) Method section: Coagulation studies INR/PT were mentioned as a test for liver synthetic function but were not provided in the results.

Prothrombin time has been added to the Results section (page 8 and page 11, Figure 4 and Supplementary Figure 2). INR was found to be non-contributory as it is derived from PT and was therefore removed.

23) Method section: Please provide a reference of an article that describes haemolysis due to rewarming. What is the evidence that the temperature should be increased with 1 degree Celsius per hour? Why start at 32 degrees and not 20 or directly at 36?

We utilised controlled rewarming to minimise ischaemia reperfusion injury to the liver. We chose 32 degrees C to avoid compromising the integrity of red blood cells which could result in haemolysis. This has been clarified in the manuscript and references added as requested (Methods, page 17).

van Leeuwen, O. B. et al. Transplantation of High-risk Donor Livers After Ex Situ Resuscitation and Assessment Using Combined Hypo- and Normothermic Machine Perfusion: A Prospective Clinical Trial. *Ann Surg* **270**, 906-914, (2019).

Hoyer, D. P. et al. Controlled oxygenated rewarming of cold stored livers prior to transplantation: first clinical application of a new concept. *Transplantation* **100**, 147-152, (2016).

24) Method section: Oxygen consumption was estimated using Henry's law; shouldn't this be in combination with Fick's law? Otherwise, it is calculated only on dissolved oxygen. Also, the formula is not complete, because it also takes into account the blood flows through the liver.

We calculated oxygen content by adding free and bound oxygen content individually for the hepatic artery, portal vein and hepatic vein: $[O_2 = pO_2 \times k + sO_2 \times Hb \times c]$ where pO_2 and sO_2 are the partial pressure of oxygen and oxygen saturations respectively, and k is the oxygen solubility coefficient and c is the oxygen carrying capacity of haemoglobin.

We then used this to calculate the overall oxygen extraction/consumption rate using the hepatic artery, portal venous and hepatic vein flow and adjusted to the weight of the livers:

$$[Oxygen\ Consumption = Flow_{HA} \times O2_{HA} + Flow_{PV} \times O2_{PV} - Flow_{HV} \times O2_{HV}].$$

These details have been added to the methods section (page 20).

25) P8, L179: "Liver and bile pH...." should be "Perfusate and bile pH..."

This has been corrected as requested.

26) Result section: AST is mentioned in the method section, but not shown in the figures.

The AST levels were not interpretable due to unreliable results and therefore non-contributory. These were therefore removed from the manuscript

27) Figure 2: Infusion of vitamins is not mentioned in the manuscript.

The details for vitamins added during perfusion have been included in the methods (page 17).

28) The authors describe a decrease in HA flow with the same HA pressure, however, this is not visible in the graph (Figure 4). It looks more like the opposite. If this is not the case, maybe the graphs are not that clear.

Hepatic artery flows did not change noticeably during perfusion until the final stages of perfusion immediately before termination of the experiment (which is difficult to appreciate in the figures). This has been corrected and clarified in the text (Results: page 9)

Reviewer #2 (Remarks to the Author):

The manuscript “Long-term ex-vivo normothermic perfusion of human split livers: a unique model to study new therapeutics and increase the number of available organs” by Dr Lau and colleagues reports development of a novel protocol and experience with a week-lasting discarded donor liver perfusions.

The authors describe series of 10 discarded donor livers, that underwent splitting during the NMP, with subsequent commencement of NMP using another device. This approach resulted in 20 extended perfusions. The manuscript is well written and presents a novel data that expanding on the work authors’ previously publications.

The extended NMP model offers tremendous potential to study resuscitation and regeneration of suboptimal donor livers. The split-organ approach in addition allows researchers to design NMP studies to test novel interventions on livers that has got an identical risk profile. Dr Lau and colleagues should be congratulated for such outstanding contribution and pushing boundaries of the liver perfusion. The manuscript represents an abundance of hard work, and the results are impressive, albeit understated. This paper makes one of the first to attempt to describe features typical of liver failure during NMP. The features described are universal but could be argued as very end-stage signs of liver failure, namely lactate >10 mmol/L and hypoglycaemia refractory to intervention. This might impact the results by categorizing organs as viable whilst their function is already progressively deteriorating.

We thank Reviewer #2 for their comments and feedback.

We recognise the impact of continued evaluation of livers after breaching ‘viability’ criteria to the point of organ failure. Importantly, all viability criteria to date are designed for use during short-term perfusion and the changes that occur in the long-term are largely undescribed. As the reviewer acknowledges, in this manuscript, we sought to describe what happens to a liver as it fails during long-term normothermic machine perfusion. This has been clarified in the text on page 7 and 19.

The authors present some features of livers able to survive 7 days, but there is a lack of discussion in why a liver fails during NMP and what separates the organs that thrive. Some organs were declined based on donor age and the NMP was commenced following a short period of cold ischaemia only (<5 hours). Could authors speculate what might be the reasons such livers’ perfusion did not exceed 168 hours? Less than 50% of the organs were able to tolerate perfusion beyond 7 days, and the overall median perfusion time was 5 days.

The reason for organ failure in the long-term is an area of ongoing research for our group. Despite some organs having a short cold time, and being declined based on age alone (which would lead an investigator to think these organs might survive a long time based on inherent quality) liver perfusion did not always exceed 1 week in these cases. We believe these organs all failed eventually due to either a missing key metabolic component required for energy homeostasis or a missing growth factor stimulus which signals hepatocytes to regenerate. This may explain why it seems difficult to achieve perfusion much longer than 7 days. This has been clarified in the discussion (page 14-15).

The main shortcoming of the paper is omission of important details that would allow readers to replicate the experiments. The aims and methodology are well defined and described, with a clear objective to explore and define the features of organs capable of surviving >7 days during NMP. However, if a reader wished to replicate the results described, there is a lack of detail in the perfusion protocol to facilitate this. Key missing aspects include formulations of drugs (& brands), flowcharts for maintaining the desired perfusate parameter levels (i.e. glucose, and haemoglobin) and these and other troubleshooting issues. As such, the statement that the presented extended NMP protocol for split human livers is both reproducible and reliable is not supported by the data included.

The technical details for our protocol have overall been enhanced such that a reader could replicate our model. The formulations and brands of drugs have been added to the methods section (page 17-18). The algorithmic flowchart for manual maintenance of perfusion parameters has been added (referenced in text page 18, Supplementary Figure 5). Details about perfusion parameters and cannulas have been added as discussed above.

Authors did not provide details about the oxygen delivery and consumption during the perfusions. Were there any changes in the perfusate quality and oxygen carrying capacity throughout the course of the perfusions? Were all perfusions performed without the need to top up the red blood cells, or replacement of blood products? What were the levels of methaemoglobin or carboxyhaemoglobin? These aspects would be worth highlighting in the Discussion.

We did not encounter issues with oxygen carrying capacity during perfusion. There was a slight decline in haemoglobin over time, we believe due to extended sampling, but this was corrected by concentration of the perfusate volume using the dialysis to maintain the levels at 55-65g/L. We believe the starting volume of 2L of perfusate carried enough redundancy such that any losses could be compensated throughout perfusion without notable deterioration in the perfusate quality. Red blood cell top ups were therefore not required. This has been clarified in the methods and discussion (page 18 and page 15). Methaemoglobin and carboxyhaemoglobin levels measured during whole liver perfusion were within normal limits and were non-contributory. Oxygen consumption was calculated as discussed above. This has been clarified in the methods section (page 20).

An important result described is the difference in hepatic arterial flow between livers that survived 7 days versus organs that didn't. The authors describe livers that were more likely to survive as having increased arterial flow. From a purely physiological perspective, hepatic arterial flow is weight dependent (about 0.25 ml/g /min). Could authors explore how the differences in flow related to liver weight or whether suitable intervention to increase flows may improve the likelihood of salvaging the failing livers? Presenting the vascular flows in relation to the liver mass (similarly to bile production) might reveal some correlation not apparent at the current data presentation form.

We recognise the importance of weight-adjustment for analysing hepatic artery flow. After adjustment for liver weight, we noted that the LLSG achieved significantly higher flows/minute/kg of liver than the ERG. This has been added to the results section (page 9), and Figure 2. The impact of high arterial flows for a small partial liver remains unclear. This also represents one of the challenges of utilising a pressure-controlled perfusion system and an adult-sized machine for a paediatric-sized organ. This has been acknowledged in the discussion (page 12).

Additionally, after adjusting hepatic artery flow by weight, the differences between livers that survived <7days and ≥7 days were still present, but less pronounced. This has been added to the results section (page 11) and Supplementary Figure 4.

The Discussion should mention which perfusion and perfusate parameters were controlled and adjusted, or which were set and not changed (hepatic artery and portal vein pressures), or which are normalised by dialysis filter (pH). This would make clearer why some graphs in the Figure 6 appear to be similar in viable and failing livers.

Acid-base balance and oxygenation were maintained manually by adjustment of oxygen-compressed air ratios and regulation of gas flows. Glucose levels were maintained by manual adjustment of glucose, insulin and glucagon infusions. Perfusate flow (hepatic artery and portal vein) was pressure controlled. This was set and not changed unless flow targets were not reached. Dialysis was manually adjusted based on potassium and haemoglobin levels. These details have been enhanced and clarified in the methods section (page 18-19). The algorithm for manual adjustment of parameters during perfusion has been included to improve clarity and allow replication of our protocol (referenced in results on page 18 and discussion on page 13) (Supplementary Figure 5).

In summary, the maintenance of 10-day normothermic liver perfusion is a tremendous success not yet achieved / published by others (and the 328 hours is the world-longest NMP liver perfusion). However, the presented data lacks detail, which limits the experiments reproducibility and manuscript informative value. The authors should provide more details to help readers to appreciate the complexity and labour-intensity of the experiments and allow to replicate them. The Discussion should comment on some aspect of extended perfusions that others might struggle to overcome. These changes would help guide others and are likely to increase the impact of this impotent paper.

The technical details for our protocol have overall been enhanced such that a reader could replicate our model as discussed above.

The major challenges encountered were perfusion of small partial livers and the labour intensity required for manual adjustment of multiple parameters. To assist others in overcoming these challenges, this has been expanded in the discussion and suggestions for future development (automation and digital control) included (page 12-13).

Minor comments

The Figure 3 shows several closely packed graphs which do not allow to inspect the data in detail

This has been reformatted to increase the size of the graphs.

The term ex situ would be preferred to ex vivo (Karangwa et al 2016)

This has been corrected throughout the text.

DCD is a widely used abbreviation for donation after circulatory death, so would be preferred to DCDD.

This has been corrected throughout the text.

If the liver is not transplanted, then it is not a graft and would be better referred to as liver / organ.

This has been corrected throughout the text.

REVIEWERS' COMMENTS

Reviewer #1 (Remarks to the Author):

Thank you for carefully revising your manuscript. I am satisfied with the modifications and clarifications and have no further comments. Congratulations on this excellent work!

Reviewer #3 (Remarks to the Author):

I've read with great interest the work from Lau et al. on "Long-term ex-situ normothermic perfusion of human split livers for more than 1 week" which described 20 prolonged ex-situ normothermic perfusion in split liver grafts in an attempt to provide valuable technical and biochemical insights on prolonged ex-situ liver graft assessment. As already mentioned, these long perfusions must have been a tremendous amount of work and great teamwork that should be acknowledged.

The authors provided an improved version of their work according to the reviewers' comments. They have indeed tried to focus on the feasibility of prolonged perfusion rather than presenting a reproducible and "universal" ex-situ perfusion strategy. This could pave the way toward an improvement of normothermic perfusion in a long-term setting.

Please, find below my comments on the manuscript:

- One main concern remains the use of viability criteria as both an endpoint for liver graft viability assessment but also as factors of "graft loss".

Beside, as previously stated, both partial liver grafts were not able to provide the same outcomes which questioned the reproducibility of the model.

I would suggest to describe this work as proof of concept study in which liver graft splitting was made as an attempt to offer two suitable liver graft for ex-situ prolonged assessment, with a particular focus on technical pitfalls of prolonged perfusion. Then, an outlook should be made on potential factors involved in "graft survival" during prolonged ex-situ

reperfusion.

The manuscript may benefit from additional corrections.

In addition, donor characteristics were not included in the results section, as potential factors impacting “graft survival”. For example, one graft with major steatosis did not reach viability criteria at 5 days (as expected?), whereas discarded grafts for logistic/subjective reasons displayed good graft function beyond 5 to 7 days.

- As stated by the authors, perfusion was pressure controlled. Portal and hepatic artery flow thereby adapt and can be analyzed as surrogates of graft compliance and quality. However, as detailed in Figure 2, liver grafts were not all perfused at either 60mmHg or 8mmHg thus perhaps explaining the low flow in some cases. Besides, the use of smaller canula (in split grafts) may lead to a higher perfusion pressure without reaching targeted flow thresholds. As in clinical practice, this could increase graft injury thus explaining graft function impairment.

Notably, hepatic artery flow was significantly increase in LLSG, which may be explained by Hepatic arterial buffer response which has also been described in ex-situ perfusion. This may be the consequence of an inadequate portal perfusion in LLSG which could explain why LLSG were less prone to “survive” beyond 7 days.

Finally, the use of the VITTAL criteria which has been described for WLG should be discussed especially regarding hemodynamic assessment.

- The authors added a more detailed evaluation of the cholangiocyte compartment in their evaluation. The manuscript may now benefit from a more detailed histological analysis based on Suzuki or the Groningen group scoring system. This may add a more objective and dynamic evaluation of histological changes during prolonged reperfusion which may be further discussed in order to better assess liver graft viability.

Suzuki S, Toledo-Pereyra LH, Rodriguez FJ, Cejalvo D. Neutrophil infiltration as an important factor in liver ischemia and reperfusion injury. Modulating effects of FK506 and cyclosporine. *Transplantation*. 1993 Jun;55(6):1265–72

Sosa RA, Zarrinpar A, Rossetti M, Lassman CR, Naini BV, Datta N, et al. Early cytokine signatures of ischemia/reperfusion injury in human orthotopic liver transplantation. *JCI Insight*. 2016 Dec 8;1(20):e89679

op den Dries S, Westerkamp AC, Karimian N, Gouw ASH, Bruinsma BG, Markmann JF, et al. Injury to peribiliary glands and vascular plexus before liver transplantation predicts formation of non-anastomotic biliary strictures. *J Hepatol*. 2014 Jun;60(6):1172–9

- The authors improved their discussion with more insights on the technical challenges of prolonged ex-situ perfusion. This remains, as stated by Hessheimer et al. a non physiological model with its inherent limitations which should be discussed.

Hessheimer, Amelia J. MD, PhD1; Vengohechea, Jordi BS1; Fondevila, Constantino MD, PhD1. Metabolomic Analysis, Perfusate Composition, and Pseudo-physiology of the Isolated Liver During Ex Situ Normothermic Machine Perfusion. *Transplantation* 107(5):p e125-e126, May 2023. | DOI: 10.1097/TP.0000000000004530

Point-by-point response to reviewer comments

We would like to thank the reviewers of *Nature Communications* for their excellent comments and feel that the paper is now stronger as a result. We have addressed all the questions raised to the best of our abilities below.

Reviewer #1 (Remarks to the Author):

Thank you for carefully revising your manuscript. I am satisfied with the modifications and clarifications and have no further comments. Congratulations on this excellent work!

We thank Reviewer #1 for all their comments and feedback.

Reviewer #3 (Remarks to the Author):

I've read with great interest the work from Lau et al. on "Long-term ex-situ normothermic perfusion of human split livers for more than 1 week" which described 20 prolonged ex-situ normothermic perfusion in split liver grafts in an attempt to provide valuable technical and biochemical insights on prolonged ex-situ liver graft assessment. As already mentioned, these long perfusions must have been a tremendous amount of work and great teamwork that should be acknowledged.

The authors provided an improved version of their work according to the reviewers' comments. They have indeed tried to focus on the feasibility of prolonged perfusion rather than presenting a reproducible and "universal" ex-situ perfusion strategy. This could pave the way toward an improvement of normothermic perfusion in a long-term setting.

Please, find below my comments on the manuscript:

- One main concern remains the use of viability criteria as both an endpoint for liver graft viability assessment but also as factors of "graft loss".

Beside, as previously stated, both partial liver grafts were not able to provide the same outcomes which questioned the reproducibility of the model.

I would suggest to describe this work as proof of concept study in which liver graft splitting was made as an attempt to offer two suitable liver graft for ex-situ prolonged assessment, with a particular focus on technical pitfalls of prolonged perfusion. Then, an outlook should be made on potential factors involved in "graft survival" during prolonged ex-situ reperfusion.

The manuscript may benefit from additional corrections.

We thank Reviewer #3 for their comments and feedback.

We recognise that our model is not mature and therefore not a truly reliable or reproducible model. The language has been softened to put forward the model as 'functional' and as a 'proof-of-concept' model as requested in the introduction (page 5) and discussion/conclusion (page 12 and page 15).

In addition, donor characteristics were not included in the results section, as potential factors impacting "graft survival". For example, one graft with major steatosis did not reach viability criteria at 5 days (as expected?), whereas discarded grafts for logistic/subjective reasons displayed good graft function beyond 5 to 7 days.

Donor factors were not significantly different between grafts that survived >7 days or ≤7days. These results are limited by the relatively small numbers of grafts but analysis of the impact of donor characteristics on organ quality was not the main focus of this study. These data have been added to the results section on page 10 and Supplementary Table 4 has been added to display the full analysis.

- As stated by the authors, perfusion was pressure controlled. Portal and hepatic artery flow thereby adapt and can be analyzed as surrogates of graft compliance and quality. However, as detailed in Figure 2, liver grafts were not all perfused at either 60mmHg or 8mmHg thus perhaps explaining the low flow in some cases. Besides, the use of smaller canula (in split grafts) may lead to a higher perfusion pressure without reaching targeted flow thresholds. As in clinical practice, this could increase graft injury thus explaining graft function impairment.

Notably, hepatic artery flow was significantly increase in LLSG, which may be explained by Hepatic arterial buffer response which has also been described in ex-situ perfusion. This may be the consequence of an inadequate portal perfusion in LLSG which could explain why LLSG were less prone to “survive” beyond 7 days. Finally, the use of the VITTAL criteria which has been described for WLG should be discussed especially regarding hemodynamic assessment.

Hepatic arterial flow was notably higher in the LLSG. We believe this is a consequence of the coeliac trunk being kept with the LLSG (larger cannula), compared to cannulation of the right hepatic artery for the ERG (smaller cannula). Similarly, the portovenous supply for the LLSG required a small cannula for the right portal vein. This is a limitation of the model, and potentially results in artificially higher or lower arterial and portovenous flows than physiological. This is a technical challenge relating to the split model and the discussion of these limitations has been expanded in the discussion section on page 12.

Regarding the haemodynamic assessment of the VITTAL criteria, all organs always met the haemodynamic criteria described in the VITTAL study. Notably, these criteria were not designed for use in partial organs, but all organs displayed haemodynamic stability until organ failure. These findings are discussed in the results section on page 9.

- The authors added a more detailed evaluation of the cholangiocyte compartment in their evaluation. The manuscript may now benefit from a more detailed histological analysis based on Suzuki or the Groningen group scoring system. This may add a more objective and dynamic evaluation of histological changes during prolonged reperfusion which may be further discussed in order to better assess liver graft viability.

Suzuki S, Toledo-Pereyra LH, Rodriguez FJ, Cejalvo D. Neutrophil infiltration as an important factor in liver ischemia and reperfusion injury. Modulating effects of FK506 and cyclosporine. Transplantation. 1993 Jun;55(6):1265–72

Sosa RA, Zarrinpar A, Rossetti M, Lassman CR, Naini BV, Datta N, et al. Early cytokine signatures of ischemia/reperfusion injury in human orthotopic liver transplantation. JCI Insight. 2016 Dec 8;1(20):e89679

op den Dries S, Westerkamp AC, Karimian N, Gouw ASH, Bruinsma BG, Markmann JF, et al. Injury to peribiliary glands and vascular plexus before liver transplantation predicts formation of non-anastomotic biliary strictures. J Hepatol. 2014 Jun;60(6):1172–9

We recognise that an in-depth analysis of the cholangiocyte compartment would be of interest in the field; particularly to understand the role of biliary regeneration in early and late biliary viability. In the current study, we aimed to establish a model of long-term ex situ perfusion of split livers with a particular focus on hepatocellular function and factors relating to organ survival. Therefore, detailed biliary evaluation could not be included within the limitations of this study. Further, detailed assessment of biliary biopsies to evaluate the cholangiocyte compartment was not possible at this time because firstly, the number of biopsies necessary (at multiple time points) for a robust assessment were not available to us and secondly, histological analysis using the Suzuki or Gronigen group scoring systems would require external validation for use in the ex-situ setting.

However, biliary viability is an area of ongoing interest and research for our group and we hope future work will provide insight into this field. This issue has been added to the discussion section as a potential area for future work on page 14.

- The authors improved their discussion with more insights on the technical challenges of prolonged ex-situ perfusion. This remains, as stated by Hessheimer et al. a non physiological model with its inherent limitations which should be discussed.

Hessheimer, Amelia J. MD, PhD1; Vengohechea, Jordi BS1; Fondevila, Constantino MD, PhD1. Metabolomic Analysis, Perfusate Composition, and Pseudo-physiology of the Isolated Liver During Ex Situ Normothermic Machine Perfusion. Transplantation 107(5):p e125-e126, May 2023. | DOI: 10.1097/TP.0000000000004530

We recognise the limitations of this model such that it provides a simulation of the physiological function of the liver, without exact replication. Notably, there is no gut-liver axis or a simulation of circadian variation. The discussion of these limitations has been expanded in the discussion section on page 15 and the reference has been added as requested.